



# 1 Tillage-induced short-term soil organic matter turnover
# 2 and respiration

**S. R. Fiedler[1], P. Leinweber[1], G. Jurasinski[1], K.-U. Eckhardt[1], S. Glatzel[1,*]**
[1]{University of Rostock, Faculty of Agricultural and Environmental Sciences,
Rostock, Germany}
[*]{now at: University of Vienna, Department of Geography and Regional Research,
Vienna, Austria}
Correspondence to: S. R. Fiedler (sebastian.fiedler@uni-rostock.de)

## 11 Abstract

Tillage induces decomposition and mineralisation of soil organic matter (SOM) by the
disruption of macroaggregates and may increase soil $CO_2$ efflux by respiration, but
these processes are not well understood at the molecular level. We sampled three
treatments (mineral fertiliser = MF, biogas digestate = BD, unfertilised control = CL) of
a stagnic luvisol a few hours before and directly after tillage, and four days later from a
harvested maize field in Northern Germany and investigated these samples by
pyrolysis-field ionization mass spectrometry (Py-FIMS) and hot-water extraction.
Before tillage, BD showed much more volatilised matter (VM) during pyrolysis,
indicating an increased amount of SOM. The Py-FIMS mass spectra revealed distinct
differences in relative ion intensities of undisturbed soil compared to BD most likely
attributable to the cattle manure used for the biogas feedstock and to relative
enrichments during anaerobic fermentation. After tillage, the $CO_2$ effluxes were
increased in all treatments, but this increase was less pronounced in BD. We explain
this by a restricted availability of labile carbon and, possibly an inhibitory effect of
sterols from digestates. Despite high spatial variability, significant changes in SOM
composition were observed following tillage. In particular, lignin decomposition and
increased proportions of N-containing compounds were detected in BD. In MF, lipid




proportions increased at the expense of carbohydrates and peptides, indicating an
enhanced microbial activity. SOM composition in CL was unaffected by tillage. In
summary, combining all analyses data provided strong evidence for significant short-
term SOM changes due to tillage in fertilized soils.

## 34    1    Introduction

The influence of tillage on soil organic matter (SOM) is generally well understood.
Tillage stimulates decomposition of SOM resulting in increased $CO_2$ efflux (Alvarez et
al., 2001; Dao, 1998; Liu et al., 2006), mostly by aeration and by the disruption of SOM
that had been protected in macro-aggregates (Grandy and Robertson, 2007; Six et al.,
1999). In the long-term, tillage promotes a shift of chemical structure and age towards
more recent SOM (Grandy and Neff, 2008) due to both, the mineralisation of older
SOM and the decomposition of recent plant residues (Balesdent et al., 1990). In
addition, tilled soils contain lower amounts of labile organic matter (Balota et al., 2003)
and have an increased potential for mineralisation and nitrification (Doran, 1980) which
implies a lower potential to immobilise mineral N (Follett, R. F. and Schimel, D. S.,
1989; Schulten and Hempfling, 1992). However, the immediate, short-term effects of
tillage events on SOM are almost unknown.
Research on short term effects of tillage on SOM has focussed largely on $CO_2$ efflux:
several studies recorded the dynamics of $CO_2$ efflux immediately after tillage (cf., Table
5 in Fiedler et al., 2015) and some basic models have been developed that describe
correlations between $CO_2$ efflux and the turnover of soil organic carbon (SOC) after
tillage by first order kinetics (La Scala et al., 2008). These correlations do not causally
explain which SOC constituents that form the majority of SOM are mineralized.
Furthermore, SOM-$CO_2$-efflux-relationships are influenced by the type of soil
amendment (Fiedler et al., 2015).
Biogas digestate is a relatively new type of soil amendment, and its long-term effects on
the reproduction of the SOM level is still under debate as recently reviewed by (Möller,
2015). Consequently, it is not clear how long-term application of biogas digestates
would alter the composition of SOM, and tillage effects on short-term SOM turnover in



biogas digestate-amended soils are almost unstudied. Even short-term changes of SOM
may have strong effects on nutrient availability and plant productivity. A better
understanding of the immediate impacts of tillage on SOM and its turnover may help to
avoid adverse effects for plant growth (Doran, 2002; Franzluebbers et al., 1994;
Mijangos et al., 2006).
In general, detecting changes in the molecular-chemical composition of SOM in time
periods as short as days, requires extremely sensitive methods. Pyrolysis-field
ionization mass spectrometry (Py-FIMS) is a very sensitive method and has been
applied successfully to investigate differences in the chemical composition of SOM
under different fertiliser treatments like mineral NPK-fertiliser or farmyard manure
(Jandl et al., 2004; Leinweber et al., 2008b; Schmidt et al., 2000). Even very small
alterations in the composition and stability of dissolved organic matter – a very reactive
part of SOM – during storage in the fridge (Schulten et al., 2008) or diurnal cycles of
$CO_2$-assimilation and respiration (Kuzyakov et al., 2003; Melnitchouck et al., 2005;
Leinweber et al., 2008a) have been detected and resolved by multivariate statistics of
mass-spectrometric fingerprints. Furthermore, Py-FIMS of bulk SOM revealed
alterations in laboratory incubation experiments and linked these to respiration and
enzyme activities (Leinweber et al., 2008b). However, it is unclear if the method is
sensitive enough to detect tillage-induced SOM alterations under various fertilisation
regimes and analyse its influence on $CO_2$ efflux at the field scale where spatial
heterogeneity may interfere with the temporal dynamics much more than in the above
cited laboratory studies.
Here, we investigate (1) short-term effects of tillage on SOM composition and (2)
potential relationships between decomposable SOM fractions and measured $CO_2$ efflux
under the impact of different soil amendments by combining Py-FIMS with $CO_2$ efflux
measurements.



## 2  Materials and methods

### 2.1  Study site

The study site is located in northeast Germany in the ground moraine of the
Weichselian glacial period at 53° 48' 35" N and 12° 4' 20" E (elevation 10 m) within a
gently rolling relief. The soil is a stagnic luvisol (IUSS Working Group WRB, 2006)
with loamy sand texture overlying bedrock of till. The top soil (0-30 cm) has an organic
carbon content of 1.16% (standard deviation (SD) = 0.1, $n$ = 3, measured with CN-
analyser "vario MAX", Elementar, Hanau, Germany), pH of 7.4 (SD = 0.9, $n$ = 3,
measured in $H_2O$ with pH meter "CX-401", Elmetron, Zabrze, Poland) and bulk density
of 1.51 g $cm^{-3}$ (SD = 0.08, $n$ = 3, measured on 250 cm³ soil cores). The climate is
characterized by maritime influence with annual averages of 8.8° C temperature and
557 mm total precipitation for the 30-year-period from 1985 until 2014 (LFA 2015).
The experiment was conducted on a field which has been cultivated with maize (*Zea*
*Mays* L.), cultivar "Atletico", as feedstock for a biogas plant. The previous crops were
winter wheat (*Triticum aestivum* L.) followed by maize.
We compared three fertiliser treatments: CL – without fertiliser (control), MF – with
mineral fertiliser, and BD – with biogas digestate. The size of the three experimental
plots was 6 by 30 m each. In both fertilised treatments, equal overall amounts of plant-
available N were applied (160 kg $ha^{-1}$) on 26 April 2012. The mineral fertiliser calcium
ammonium nitrate was top-dressed whereas the biogas digestate was injected into the
soil down to 10 cm depth with a track width of 25 cm. Following the research facility
for agriculture and fisheries (LFA) of the federal state of Mecklenburg-Western
Pomerania, Germany (personal communication, 2014), a mineral fertiliser equivalent of
70% of total N in the biogas digestates (229 kg N $ha^{-1}$) was assumed. The biogas
digestate originated from the anaerobic fermentation of 91% cattle slurry, 7% rye groats
and 2% maize silage; it had  pH 8.1, and 3.8% C, 0.5% total N and 0.3% $NH_4$-N in
original matter.
Sixteen days after harvest of the maize (8 October 2012), the field site was first tilled
with a disc harrow "Väderstad Carrier 300" down to 10 cm depth (24 October, about



9.15 a.m.) and then with a reversible plough "Överum CX 490" down to 30 cm depth
on the subsequent day (25 October, about 11.30 a.m.).
**2.2   $CO_2$ concentration measurement and estimation of $CO_2$ efflux**
For measuring $CO_2$ exchange we permanently installed three replicate collars in each
treatment after fertilisation in spring which were removed for tillage and inserted back
afterwards. The adjacent collars shared distances of 1m. The collars had a total height of
15 cm and were installed into the soil down to 12 cm depth. The $CO_2$ concentration
measurements where performed with two LI-COR (Inc., Lincoln, NE, USA) LI-820
infrared gas analysers, each connected to a non-steady state closed chamber that was
placed on the collars during measurements. The chambers had a square area of 0.62 m²
and a height of 0.55 m, resulting in a chamber volume of 0.34 m³ and were equipped
with small fans (80 x 80 x 25 mm, 3000 rpm, 68 m³ h$^{-1}$) in order to mix and homogenize
the air inside the chambers. Due to the successive measurement of the replicates in each
treatment, we obtained pseudo-replications.
During chamber placement, we recorded $CO_2$ concentrations in the chamber headspace
with 1.3 s intervals for 3 to 5 min, resulting in approximately 140 to 230 data points per
measurement. Fluxes were estimated with function *fluxx* of package *flux* version 0.3-0
(Jurasinski et al., 2014) for the R statistical software version 2.15.2 (R Core Team,
2013). In short, the algorithm identifies the most linear part of the $CO_2$ concentration
development during chamber placement time and fits a linear regression model (Eq.

135    (1)):

$$f = \frac{MpV}{RTA}\frac{dc}{dt}10^6,  \qquad (1)$$
with $f$ the $CO_2$ flux (g m$^{-2}$ h$^{-1}$), $M$ the molar mass of $CO_2$ (44 g mol$^{-1}$), $p$ the air pressure
(Pa), $V$ the chamber volume (m$^3$), $R$ the gas constant (8.31 J mol$^{-1}$ K$^{-1}$), $T$ the
temperature inside the chamber (K), $A$ the area covered by the chamber (m$^2$), and $dc/dt$
the $CO_2$ concentration change over time (ppm h$^{-1}$). The minimum proportion of data
points to be kept for regression analyses was 70 % of a concentration measurement to
discard data noise at the beginning and the end resulting from chamber deployment and



removal (for details see help file for function *fluxx* of package *flux*). Thus, each $CO_2$
flux was estimated at least from 98 concentration measurements. Only linear fluxes
with a concentration change of at least 10 ppm, a normalised root mean square error
(NRMSE) $\leq 0.15$ and a coefficient of determination ($R^2$) of at least 0.85 were included
in further analyses. We assumed linearity of concentration change and did not test for
non-linearity since 95.1% of the obtained linear regressions had $R^2 \geq 0.95$.
To obtain reference data from before tillage operations, the undisturbed site was
measured hourly between 7 a.m. and 1 p.m. on 19 October 2012 (i.e. between harvest
and tillage). The intervals between measurements before, during and after tillage
operations were varied to effectively capture the development of $CO_2$. The
measurements immediately after the tillage operations were conducted within one
minute by inserting the collars and putting on the airtight chambers. The timeline (24
till 29 October) of tillage events, soil samplings and the respective $CO_2$ measurements,
together with soil temperature, is shown in Fig. 1. After this period, $CO_2$ measurements
were performed hourly before noon on 1, 5 and 9 November.

### 2.3  Soil sampling and analyses

Three replicates of bulk soil samples were taken at 5 – 15 cm depth with soil sample
rings (V = 250 cm³) in a triangular arrangement between the three collars for gas
sampling (see 2.3) in each treatment at three dates: 1) right before the first tillage
operation, 2) in the afternoon after the second tillage operation and 3) four days after the
second tillage operation. The resulting 27 soil samples were fixed immediately with
liquid nitrogen and splitted thereafter into subsamples for freeze-drying and for oven-
drying at 60° C.
For Pyrolysis-field ionization mass spectrometry (Py-FIMS), about 5 milligrams of the
freeze-dried, ground and homogenized samples were thermally degraded in the ion
source (emitter: 4.7 kV, counter electrode -5.5 kV) of a double-focusing Finnigan MAT
mass spectrometer (Finnigan, Bremen, Gemany). The samples were heated in a
vacuum of $10^{-4}$ Pa from 50 °C to 700 °C, in temperature steps of 10 °C over a time
period of 15 minutes. Between magnetic scans the emitter was flash heated to avoid



residues of pyrolysis products. The Py-FIMS mass spectra of each sample were gained
by the integration of 65 single scans in a mass range of 15 – 900 *m/z*. Ion intensities
were referred to 1 mg of the sample. Volatile matter was calculated as mass loss in
percentage of sample weight. The three replicates of each sample were then averaged to
one final survey spectrum. Moreover, thermograms were compiled for the total ion
intensities. The assignment of marker signals to chemical compounds from the survey
spectra were interpreted according to (Leinweber et al., 2013) to obtain the relative
abundance of ten SOM compound classes: 1) carbohydrates, 2) phenols and lignin
monomers, 3) lignin dimers, 4) lipids, alkanes, alkenes, bound fatty acids and alkyl
monoesters, 5) alkylaromatics, 6) mainly heterocyclic N-containing compounds, 7)
sterols, 8) peptides, 9) suberin, and 10) free fatty acids.
Subsamples of oven-dried and sieved soil (2 mm) were used for determination of total
and hot water-extracted C and N. For determination of total C and N, 1 g of ground soil
was analysed with a vario Max CN Element Analyzer (elementar Analysensysteme
GmbH, Hanau, Germany) based on high temperature combustion at up to 1200 °C with
subsequent gas analysis. For hot-water extraction, 20 g soil were boiled in 40 ml
deionized water for 60 minutes (Leinweber et al., 1995). After filtration with pleated
filter (240 mm, 80 g m$^{-2}$) by Munktell (Falun, Sweden), extracts were analysed with a
DIMATOC 2000 (DIMATEC Analysentechnik GmbH, Essen, Germany) for
determination of hot-water extractable organic C (HWC) and total nitrogen bound
(HWN). These measurements of organic C and total nitrogen bound are based on the
principle of thermal-catalytic oxidation with subsequent NDIR detection and the
principle of chemiluminescence, respectively. For each sample, two replicates were
analysed and results were averaged for further calculations.
**2.4   Statistical analyses**
All statistical analyses were run using R 2.15.2 (R Core Team, 2013). The cumulated
$CO_2$ effluxes were estimated by a bootstrap method with the function *auc.mc* of the R
package *flux* version 0.3-0 (Jurasinski et al., 2014). In detail, the $CO_2$ fluxes were
cumulated in 250 iterations, while for each run 25 fluxes were omitted randomly for the
period after tillage. For the reference period before tillage, in each iteration run 4 fluxes





were omitted randomly. The numbers of randomly omitted fluxes per run correspond
roughly to one fifth of the recorded fluxes per treatment in the respective periods. The
resulting data were used to calculate means and standard deviations. Tukey's HSD test
was applied to test for differences in means of $CO_2$ fluxes as well as of HWC and HWN
between sampling periods and treatments against a significance level of $\alpha < 0.05$. Py-
FIMS signals of the compound classes were tested for differences in means by Tukey's
HSD test against a significance level of $\alpha < 0.1$ since the number of replicates was
limited and the variances rather high. A principal component analysis was applied to the
mass signals with significant differences between the samples according to univariate
Wilk's $\lambda$ ($p < 0.001$) with function *rda* of R package vegan version 2.3-0 (Oksanen et
al., 2015).

**3    Results**
**3.1  Soil organic carbon, nitrogen, hot-water extractable carbon and hot-**
**water extractable nitrogen**
Before tillage, the soil of all treatments had similar C and HWC contents, while the N
and HWN contents were slightly higher in MF, resulting in significantly narrower C/N
and HWC/HWN ratios in MF (8.55 and 5.93, respectively) compared to BD (9.03 and
8.54, respectively) (Table 1). The C, N and HWC contents of all treatments were
changed only slightly by tillage, but the HWN content of soil in BD increased from 0.05
mg g$^{-1}$ (5.6 % of N) up to 0.07 mg g$^{-1}$ (7.4 % of N), resulting in a significant ($p < 0.05$)
narrowing of the HWC/HWN ratio from 8.5 down to 6.0 (Table 1).
**3.2  Soil $CO_2$ efflux**
Five days before the tillage operations (19 October 2012), the mean efflux rates (all in g
$CO_2$-C m$^{-2}$ h$^{-1}$) were 0.133 (CL), 0.192 (MF) and 0.173 (BD), with the efflux being
significantly lower from CL than from the amended plots MF and BD ($p < 0.05$) (Fig.
2). In the morning before the first tillage operation with a disc harrow (24 October), the
effluxes had similar magnitudes and proportions like five days before (CL = 0.147, MF



= BD = 0.199, all in g $CO_2$-C $m^{-2}$ $h^{-1}$). After harrowing, $CO_2$-effluxes increased to 0.849
(CL), 0.833 (MF) and 0.479 (BD). Over the next 5.5 hours, these values declined to
0.602 (CL), 0.460 (MF) and 0.276 (BD) resulting in overall mean effluxes of 0.554
(CL), 0.481 (MF) and 0.344 (BD), with the latter being now significantly lower
($p < 0.05$) than CL or MF during the measured period after harrowing. Directly before
the second tillage operation with a reversible plough in the morning of the following
day (25 October), the mean effluxes were 0.299 (CL), 0.249 (MF) and 0.290 (BD) (all
in g $CO_2$-C $m^{-2}$ $h^{-1}$). Immediately after ploughing, they increased sharply up to 2.443
(CL), 2.654 (MF) and 3,347 (BD) and declined to 0.371 (CL), 0.718 (MF) and 0.223
(BD) after 4 hours, leading to overall mean effluxes of the measured period after
ploughing of CL = 1.012, MF = 1.392, and BD = 1.020. Although the mean $CO_2$ fluxes
within each treatment differed significantly ($p < 0.05$) from the other measured days
only after ploughing (25 October), BD on average showed significantly ($p < 0.05$) lower
fluxes than CL or MF after tillage on 24 and 29 October (Fig. 3) as well as on 1
November (CL = 0.262, MF = 0.242, BD = 0.113, all in g $CO_2$-C $m^{-2}$ $h^{-1}$) and 5
November (CL = 0.331, MF = 0.316, BD = 0.074, all in g $CO_2$-C $m^{-2}$ $h^{-1}$).
**3.3   Pyrolysis-Field Ionisation Mass Spectroscopy**
The thermograms of total ion intensity (TII) and the Py-FIMS mass spectra of the soil
samples of CL and MF taken before tillage were similar whereas the ones of BD were
different from those two (Fig. 4): The TII-thermograms of CL and MF had a peak at
480 °C, but BD displayed a pronounced bimodal shape with a first volatilisation
maximum at about 390 °C which was less marked in CL and MF. Furthermore, the
mass spectrum of BD differed distinctly from the mass spectra of MF and CL,
especially in the abundance of marker signals for carbohydrates and peptides (e.g., *m/z*
58, 60, 84, 69, 110, 126 and 162). Apart from this the spectra are dominated by signals
for lignin mono- and dimers (e.g., *m/z* 150, 208, 222, 244) as well as for homologous
series of alkenes and alkadienes from *n*-$C_{18}$ up (e.g., m/z 252, 264/266, 278/280, 294,
308, 322, 336, 364, 392, 406) (Fig. 4).
After discriminant function analysis with Wilk's $\lambda$, the resulting significant relative
mass signals ($p < 0.001$, $n = 67$) were further explored by PCA. The first two principal





components explained 78.3% and 8.3% of total variance. All treatments are well
separated from each other (Fig. 5), with CL mainly in the 3rd quadrant, MF mainly in
the 1st and BD spanning from the 2nd to the 4th quadrant. According to this analysis,
samples from MF and BD taken before the tillage events (pre-) showed the largest
differences in composition. The PCA separated the samples taken at different dates
(pre-, post- and post + 4) in the treatments MF and BD but not in CL.
Basic data of the Py-FI mass spectra and the proportions of compound classes are
compiled in Table 2. Approximately 46.9% of the TII in the mass spectra could be
explained by m/z signals assigned to the compound classes. Additionally, non-specific
low-mass signals and isotope peaks contributed 2.6% and 14.2%, respectively. Before
tillage, VM was highest in BD although the differences in means were not significant ($p$
$> 0.1$). However, four days after tillage VM increased to 7.1% in BD and then
significantly ($p < 0.05$) exceeded that in MF and CL. Such an increase over time was
only observed for BD, but it was not significant ($p < 0.01$). In the other treatments, a
temporal increase in VM occurred directly after the first tillage with disc harrow.
The relative (Table 2) and absolute (data not shown) ion intensities of the compound
classes varied across treatments before tillage and changed differently after tillage. In
the undisturbed soil, BD had the lowest proportions of carbohydrates, heterocyclic N-
containing compounds and peptides and the highest proportions of lignin dimers, lipids,
sterols, suberin and free fatty acids. CL was characterized by higher proportions of
phenols and lignin monomers whereas MF ranged between BD and CL regarding the
proportions of these compound classes. In BD, the relative signal intensities of the
samples taken after tillage displayed significant ($p < 0.1$) increases of carbohydrates,
phenols and lignin monomers, alkylaromatics, heterocyclic N-containing compounds
and peptides while lignin dimers, lipids, sterols and free fatty acids decreased. In MF,
the proportion of lipids increased while carbohydrates and peptides decreased. No
changes were detected in the unfertilised treatment CL.
Linear correlations were calculated to investigate relationships between HWC, HWN
and soil respiration as suitable indicators of SOM dynamics (Kuzyakov, 2006;
Leinweber et al., 1995) and the absolute signal counts of the compound classes (Fig. 6).



The latter was derived from Table 2 by Eq. (2).
$$CII_{abs} = \frac{TII \times CII_{rel}}{100},$$    (2)
with $CII_{abs}$ the absolute ion intensity of the respective compound class, TII the total ion
intensity and $CII_{rel}$ the proportion of the ion intensity of the respective compound class.
In MF only the ion intensities for carbohydrates were positively correlated with HWC
whereas in BD more compound classes correlated with the tested indicators of SOM
dynamics. Here, HWC was positively correlated with the ion intensities of lignin
dimers, lipids, alkylaromatics, sterols and suberin, but no such correlation was found for
carbohydrates in disagreement to MF. However, HWN showed a positive correlation
with carbohydrates in BD. HWN was also positively correlated to phenols, lignin
monomers and heterocyclic N-containing compounds but negatively correlated to free
fatty acids. $CO_2$ efflux increased with decreasing amounts of sterols and suberin in BD.

**4    Discussion**
**4.1    Bulk soil and hot-water extracted carbon and nitrogen**
The C- and HWC-contents of the treatments showed no significant differences before
tillage (Tab. 1). However, the observed higher N- and HWN-contents in MF (Tab. 1)
did not confirm the outcomes of other experiments with similar fertilisers. No
significant differences in soil C and N were found between MF and BR in the field
(Odlare et al., 2014). On the contrary, in a pot experiment with maize by (Bachmann et
al., 2011), the soil N content was higher under application of biogas digestate compared
to application of mineral fertiliser. Since the latter and the present study were rather
short-termed (weeks and months, respectively), the C- and N-contents obtained may not
be representative for long-term effects of mineral fertiliser vs. biogas digestates.
The increase in HWN in BD after tillage indicates an increase of easily mineralisable
organic N which probably originates from soil biomass and lysates (Ghani et al., 2003;
Leinweber et al., 1995; Raich and Potter, 1995) and implies an accelerated microbial



turnover of soil organic N. This seems reasonable since the microbial community is able
to adjust its structure and activity relatively fast to utilise formerly protected organic
matter after exposure due to disruption of aggregates by tillage (Jackson et al., 2003; La
Scala et al., 2008; Mueller et al., 2014). Accordingly, (Schulten et al., 1997) observed a
short-lived increase of HWC after the first of two days of several tillage operations
which was not found in the present study. Most likely, we just did not detect it, because
we took no soil samples after the first day. Overall, a single amendment with biogas
digestates very likely is insufficient to initiate changes in bulk soil C- and N-levels.
However, the increased HWN-levels in BD can be ascribed to a tillage promoted
microbial turnover of soil organic N, confirming that the hot water extracts are a
particularly sensitive approach to detect early SOM changes (Haynes, 2005).

### 4.2   Soil $CO_2$ efflux

The immediate and sharp increase of $CO_2$ efflux from soils just after tillage is a well-
documented response and seems to be mainly driven by the release of trapped $CO_2$ from
broken up aggregates by tillage (Calderon and Jackson, 2002; Ellert and Janzen, 1999;
Reicosky et al., 1997). It is commonly suggested that a few hours afterwards, waning of
this physical outgassing is accompanied by an increased soil respiration due to a better
substrate supply for microorganisms from disrupted aggregates as well as increased soil
aeration (Schulten et al., 1997; Grandy and Robertson, 2007). The amounts of the
observed fluxes are well in accordance with the findings of previous studies, e. g.,
(Rochette and Angers, 1999) and can be explained both by the magnitude of the
disturbance, i.e. soil comminution, and the fertilisation history of the soil (Schulten et
al., 1997).
The smaller relative efflux from BD compared to MF and CL after tillage is remarkable
since before tillage the $CO_2$ fluxes in BD were of the same magnitude as those in MF
and exceeded those in CL (Fig. 2). This becomes particularly evident when one
considers the relation of cumulated $CO_2$ fluxes between the treatments before (19
October) and after tillage (24 – 29 October) (Fig. 3). Before tillage, the ratio of
cumulated $CO_2$ fluxes in $CL : MF : BD$ was $1 : 1.27 : 1.21$ and changed to
$1 : 1.21 : 0.71$ after tillage. The relatively lower $CO_2$ efflux from BD after tillage may



have different reasons. On the one hand, the organic matter originating from the
digestates is likely less available to soil microorganisms, i. e. more "recalcitrant", since
the most labile C has been consumed already in the biogas reactor (Thomsen et al.,
2013; Möller, 2015; Wentzel et al., 2015). As a consequence, the effect of increased
$CO_2$ efflux after tillage as observed in CL and MF, may have been substantially reduced
by a relative shortage of labile substrate in BD that affects the above suggested
increased soil respiration due to substrate supply after tillage. On the other hand, the
narrower HWC/HWN ratio in BD after tillage suggests an improved N supply for soil
microbes which might have enhanced their C use efficiency. Such an enhanced C use
efficiency may be accompanied by decreased C losses to heterotrophic respiration as
long as C availability is not limited (Schnitzer, 2001; Sinsabaugh et al., 2013).
However, N addition decreased the respiration when C was limited in laboratory
incubation experiments (Eberwein et al., 2015). Furthermore, (Oades, 1984) observed
decreasing $CO_2$ fluxes from soil under N saturating conditions and dextrose
amendments of 1.5 and 3 mg $g^{-1}$ soil in comparison to non-saturating conditions, but
increased $CO_2$ fluxes after dextrose amendments $\geq 7.5$ mg $g^{-1}$ soil. This supports the
assumption of not limited, but rather low levels of available C in the soil of BD. Also
the proportion of carbohydrates in BD derived from Py-FIMS, as discussed below,
consolidates this assumption.
**4.3 Pyrolysis-Field Ionisation Mass Spectroscopy and synthesis**
Generally, the Py-FIMS basic data and mass spectra (Fig. 4) and the proportions of
compound classes (Tab. 2) confirm published data from this method for Luvisols in
terms of relatively high shares of lignin monomers, phenols and alkylaromatics
(Leinweber et al., 2009). Lignin monomers and phenols might be complementarily
attributed to residues of the just harvested maize. Indeed, (Gregorich et al., 1996) found
that these are important components of maize leaves and roots as well as the light
fraction of the soil under this crop. However, the Py-FIMS data indicate differences in
SOM composition between the fertilization treatments and a pronounced impact of
tillage in the treatments MF and BD.



In the spectra of samples from BD, the additional peak at 390° C in the TII-thermogram
(Fig. 4) can be attributed mainly to phenols and lignin monomers which likely
originated from primary organic matter residues since this relatively low volatilization
temperature indicates labile and fairly undecomposed organic matter (Leifeld and
Lützow, 2014; Ludwig et al., 2015; Sleutel et al., 2011). It is reasonable to refer this
organic matter to residues from the application of BD. The VM, which is an indicator of
the SOM content (Sorge et al., 1993; Wilcken et al., 1997) but also of its stability
(Ludwig et al., 2015; Leinweber and Schulten, 1995), was larger in BD before tillage
than in MF and CL. This suggests a tendency to elevated SOM due to application of
rather stable organic matter with biogas digestate. Its increase after tillage might be
explained by a general destabilization, perhaps by an enhanced SOM turnover due to an
improved microbial accessibility to relatively recalcitrant residues of BD after tillage
(Dao, 1998; Dungait, J. A. J. et al., 2012). The temporal increase in VM directly after
the first tillage with disc harrow in MF and CL may indicate a similar increased
accessibility of SOM. But here, the newly available SOM has been depleted quickly by
microbial respiration since the microbial community is able to respond rapidly to
disturbances of arable soils (Jackson et al., 2003). This assumption is supported by the
decreasing shares of carbohydrates in MF.
The compound classes of BD revealed the largest proportions of lignin dimers, lipids,
sterols, suberin and free fatty acids at the expense of carbohydrates, heterocyclic N-
containing compounds and peptides before tillage (Tab. 2). Such a SOM composition
most likely reflects the cattle manure and plant residues of the biogas feedstock and
their relative depletions (amides and polysaccharides) or enrichments (lignins and long-
chain aliphatic compounds) during anaerobic fermentation (Leinweber et al., 1992;
Möller, 2015; van Bochove et al., 1996). The pronounced tillage effect in this treatment,
obvious from the increased relative signal intensities of carbohydrates, phenols and
lignin monomers, alkylaromatics, heterocyclic N-containing compounds and peptides at
the expense of lignin dimers, lipids, sterols and free fatty acids following tillage (Tab.
2), suggests the decomposition of lignin and the new formation of carbohydrates and
peptides. This is in line with reports of a lignin decomposition faster than that of the
total SOM (Leinweber et al., 2008b; Rasse et al., 2006; Thevenot et al., 2010). (Kalbitz



et al., 2003) suggested that lignin-derived moieties and lipids are utilised by
microorganisms at low initial availability of carbohydrates, accompanied by an
accumulation of the resulting microbial metabolites like carbohydrates and peptides.
Our data from the BD treatment supports this suggestion. Furthermore, the built up of
heterocyclic N-containing compounds might also imply a relative shortage of available
carbohydrates during the microbial transformation (Follett, R. F. and Schimel, D. S.,
1989; Gillespie et al., 2014; Schulten and Hempfling, 1992). The increased proportion
of lipids at the expense of carbohydrates and peptides in MF likely results from
increased heterotrophic respiration of labile substrates driven by enhanced microbial
activity after tillage (La Scala et al., 2008; Reicosky and Archer, 2007; Zakharova et al.,
2014). The minor changes in SOM compounds in CL might be a consequence of the
wider HWC/HWN ratio compared with MF since a lack of available N is known to
decrease the efficiency of microbial activity (Schnitzer, 2001; Sinsabaugh et al., 2013).
The positive linear correlation of HWC with lignin dimers, lipids, alkylaromatics,
sterols and suberin in BD (Fig. 6) indicates a reasonable linkage between the dynamic
organic C fraction (as indicated by HWC) and the quantity of applied biogas digestate
(as indicated by lignin dimers, lipids, alkylaromatics, sterols and suberin). At the same
time, the microorganisms in BD may have been short in available labile C since there
was no significant ($p > 0.5$) correlation between HWC and carbohydrates. In contrast, a
significant and positive correlation was observed between HWC and carbohydrates in
MF (Fig. 6). This linkage was previously described by (Leinweber et al., 1995) and
attributed to microbial biomass (Ghani et al., 2003) and labile soil C (Sparling et al.,

1998).

Interestingly, HWN correlated positively with carbohydrates in BD. Since the major
part of carbohydrates in soils originate from microorganisms and their residues (Gunina
and Kuzyakov, 2015), this may suggest a metabolic coupling between carbohydrates
and HWN because many N-cycling processes are mediated microbially (Isobe and
Ohte, 2014). This idea is supported by the negative correlation between HWN and free
fatty acids that also hints to a coupling of the dynamic N pool with microbial activity in
BD. Actually, free fatty acids are known as a major carbon source during nitrogen
immobilisation by microbial anabolism (Kirchmann and Lundvall, 1993).



In BD, the cumulated $CO_2$ efflux and the amounts of sterols were negatively correlated
(Fig. 6). This supports the suggestion of (Heumann et al., 2011) and (Heumann et al.,
2013) that sterols may have an inhibitory effect on microorganisms of the N cycle.
Furthermore, (Negassa et al., 2011) reported a significant inhibition of the urease
activity with increasing sterol proportions in agro-industrial byproducts. Since microbial
activity can affect heterotrophic soil respiration (Ryan and Law, 2005), it is likely that
increased amounts of sterols as they are typically found in biogas digestates
(Leinweber, 2015, unpublished Py-FIMS data) delay the decomposition and, thus, may
slow down soil respiration. However, since the amounts of sterols decreased
significantly after tillage in DB (Table 2), the actual sterol contribution to reduced $CO_2$-
efflux in BD relative to the other treatments cannot be ascertained by the present data
set. In light of the contradicting observation of increased labile N after tillage in BD,
inhibitory effects of sterols as reported in the above publications may be more
pronounced in undisturbed soils.
Our data and analyses suggest a short-term induction of an enhanced microbial N-
turnover by tillage under fertilisation with biogas digestates. This is supported by the
results of each of the used methods and their cross-validation, i.e., (i) HWN as an
indicator for labile N increased, (ii) $CO_2$ efflux as an indicator for carbon use efficiency
in terms of improved microbial N-availability decreased, (iii) Py-FIMS data pointing at
an increase of N-containing compounds along with the decomposition of lignins, and
finally, (iv) significant correlations among data sets from these methods (Fig. 6).
In MF, the depletion of HWC was linked to decreasing amounts of carbohydrates,
certainly due to increased microbial respiration, though no significant correlation with
$CO_2$ efflux was found. No modifications were detected in CL were the absence of
amendment may have led to a shortage of N as indicated by the relatively high
HWC/HWN-ratio which likely inhibited an enhanced microbial activity.

**5   Conclusions**
Combining Py-FIMS as a sensitive technique to detect differences and alterations of
specific compound classes of SOM with classical methods like hot-water extraction and



measurements of soil $CO_2$ efflux allowed us to gain a better understanding of short-term
SOM turnover after tillage operations. After tillage, SOM composition changed in the
temporal scale of days and the changes varied significantly under different types of
amendment. Particularly obvious were the turnover of lignin-derived substances and the
depletion of carbohydrates due to soil respiration. Thus, in BD, the SOM turnover was
relatively fast, questioning the suggested recalcitrance of biogas digestates as stable
leftovers of the anaerobic fermentation. Since we found indications for inhibitory
effects of sterols on the $CO_2$ efflux, which were previously reported in three
independent studies on parameters of the N-cycle, their long-term impact on SOM
stocks should be examined more closely. Therefore, future investigations should
address the short- and long-term turnover of SOM following various soil amendments,
especially with the relatively new biogas digestates.

**Acknowledgements**
We thank the technicians Steffen Kaufmane and Sascha Tittmar for their assistance in
field work and the research facility for agriculture and fisheries of the federal state of
Mecklenburg-Western Pomerania (LFA) in Gülzow for their co-operation, especially
Jana Peters and Andreas Gurgel. The joint research project underlying this report was
funded by the German Federal Ministry of Food and Agriculture (funding identifier
22007910).  Py-FIMS analyses in the Mass Spectrometry Laboratory of Soil Science
were funded by the "Exzellenzförderprogramm" of the Ministry of Education, Science
and Culture, federal state of Mecklenburg-Vorpommern (Project UR 07 079) as well as
by the German Federal Ministry of Food and Agriculture (funding identifier 22400112).



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



Table 1. Means and standard deviations of soil organic carbon (C), nitrogen (N), C/N ratio, hot-water extractable carbon (HWC) and
nitrogen (HWN) and HWC/HWN ratio before and after tillage. Different letters in each column indicate significant differences ($p < 0.05$)
in means. Additionally, significant changes within a treatment (BD, biogas digestate; MF, mineral fertiliser; CL, control) are highlighted in
bold.

| Treatment | Date | C (mg g⁻¹) | N (mg g⁻¹) | C/N | HWC (mg g⁻¹) | HWN (mg g⁻¹) | HWC/HWN |
|---|---|---|---|---|---|---|---|
| BD | Pre | 8.4 ± 0.0 | 0.9 ± 0.0 [b] | 9.0 ± 0.1 [a] | 0.44 ± 0.02 | 0.05 ± 0.00 [bc] | **8.5 ± 0.1 [a]** |
|  | Post | 8.5 ± 0.1 | 1.0 ± 0.0 [ab] | 8.8 ± 0.3 [ab] | 0.44 ± 0.03 | 0.07 ± 0.01 [ab] | **6.1 ± 0.4 [b]** |
|  | Post + 4 | 8.4 ± 0.0 | 1.0 ± 0.0 [ab] | 8.7 ± 0 [ab] | 0.40 ± 0.02 | 0.07 ± 0.01 [abc] | **6.0 ± 0.4 [b]** |
| MF | Pre | 8.7 ± 0.3 | 1.0 ± 0.0 [a] | 8.5 ± 0.2 [b] | 0.44 ± 0.05 | 0.08 ± 0.00 [ab] | 5.9 ± 0.8 [b] |
|  | Post | 8.5 ± 0.3 | 1.0 ± 0.0 [ab] | 8.5 ± 0.1 [b] | 0.42 ± 0.04 | 0.09 ± 0.02 [a] | 4.9 ± 0.7 [b] |
|  | Post + 4 | 8.6 ± 0.3 | 1.0 ± 0.0 [a] | 8.5 ± 0.2 [b] | 0.39 ± 0.00 | 0.07 ± 0.01 [abc] | 5.5 ± 0.5 [b] |
| CL | Pre | 8.5 ± 0.2 | 1.0 ± 0.0 [ab] | 8.8 ± 0.2 [ab] | 0.50 ± 0.10 | 0.06 ± 0.02 [abc] | 8.9 ± 1.3 [a] |
|  | Post | 8.6 ± 0.2 | 1.0 ± 0.0 [ab] | 8.8 ± 0 [ab] | 0.48 ± 0.04 | 0.06 ± 0.01 [bc] | 8.8 ± 0.8 [a] |
|  | Post + 4 | 8.5 ± 0.0 | 1.0 ± 0.0 [ab] | 8.7 ± 0.1 [ab] | 0.40 ± 0.03 | 0.04 ± 0.00 [c] | 9.6 ± 0.3 [a] |






Table 2. Total ion intensity (TII), percentage of matter volatilised in pyrolysis (VM), and relative contribution of soil organic matter compound classes to the TII as detected by Py-FIMS in the treatments (CL, control; MF, mineral fertiliser; BD, biogas digestate) close before (Pre) and after tillage (Post) and also four days after tillage (Post + 4) with standard deviations. Different letters in a column of each treatment indicate significant ($p < 0.1$) differences in means of the different dates towards tillage. Additionally, treatments with significant changes are highlighted in bold.

| Treatment | Date | TII ($10^6$ counts mg$^{-1}$) | VM (%) | Relative proportions of compound classes (% TII)* | | | | | | | | | | |
|---|---|---|---|---|---|---|---|---|---|---|---|---|---|---|
| | | | | CHYDR | PHLM | LDIM | LIPID | ALKYL | NCOMP | STEROL | PEPTI | SUBER | FATTY | Sum |
| BD | Pre | 44.3 ± 11.5 | 5.2 ± 1.3 | **3.7 ± 1.8** [a] | **9.8 ± 3.8** [a] | 3.4 ± 1.4 | **5.3 ± 1.0** [a] | 11.9 ± 1.2 | **1.8 ± 0.8** [a] | **1.6 ± 0.7** [a] | **4.3 ± 1.2** [a] | 0.1 ± 0.1 [a] | **0.5 ± 0.2** [a] | **42.3 ± 5.4** [a] |
| | Post | 40.3 ± 19.3 | 4.7 ± 1.3 | **5.6 ± 0.3** [ab] | **13.3 ± 0.8** [ab] | 2.5 ± 0.4 | **4.1 ± 0.1** [b] | 12.5 ± 0.7 | **2.8 ± 0.2** [b] | **0.7 ± 0.2** [b] | **5.5 ± 0.3** [ab] | 0 ± 0.1 | **0.2 ± 0.1** [b] | **47.3 ± 0.9** [ab] |
| | Post + 4 | 35.1 ± 3.0 | 7.1 ± 1.2 | **6.2 ± 0.3** [b] | **14.4 ± 0.3** [b] | 1.9 ± 0.2 | **3.9 ± 0.1** [b] | 13.2 ± 0.1 | **3.2 ± 0.2** [b] | **0.6 ± 0** [b] | **5.9 ± 0.2** [b] | 0 ± 0 | **0.2 ± 0** [b] | **49.4 ± 0.7** [b] |
| MF | Pre | 34.2 ± 3.4 | 3.9 ± 1.1 | 5.6 ± 0.9 | 11.4 ± 0.7 | 2.9 ± 0.4 | **4.6 ± 0.4** [a] | 12.2 ± 0.9 | 2.7 ± 0.2 | 1 ± 0.4 | 5.4 ± 0.7 | 0 ± 0 | 0.3 ± 0.3 | 46.0 ± 0.3 |
| | Post | 39.1 ± 5.2 | 4.6 ± 1.0 | 4.6 ± 0.2 | 10.5 ± 0.6 | 3.5 ± 0.2 | **5.1 ± 0.1** [ab] | 12.4 ± 0.3 | 2.3 ± 0.1 | 1.2 ± 0.2 | 4.8 ± 0.2 | 0 ± 0 | 0.1 ± 0.1 | 44.5 ± 0.8 |
| | Post + 4 | 46.5 ± 15.8 | 4.2 ± 0.5 | 4.3 ± 1.0 | 10.3 ± 1.6 | 3.3 ± 0.5 | **5.4 ± 0.4** [b] | 12.6 ± 0.5 | 2.2 ± 0.5 | 1.2 ± 0.3 | 4.5 ± 0.4 | 0 ± 0.1 | 0.3 ± 0.1 | 44.2 ± 2.8 |
| CL | Pre | 41.5 ± 15.5 | **3.6 ± 0.6** [a] | 5.5 ± 0.3 | 14.3 ± 0.4 | 2.2 ± 0.8 | 4.3 ± 0.1 | 13.6 ± 0.4 | 3.1 ± 0.2 | 0.6 ± 0 | 5.4 ± 0.2 | 0 ± 0 | 0.2 ± 0.2 | 49.2 ± 0.9 |
| | Post | 41.2 ± 7.8 | **4.7 ± 0.4** [b] | 5.6 ± 0.3 | 14.4 ± 0.2 | 1.8 ± 0.1 | 4.5 ± 0.2 | 13.9 ± 0.1 | 3.1 ± 0.1 | 0.6 ± 0.1 | 5.4 ± 0.3 | 0 ± 0 | 0.3 ± 0.1 | 49.6 ± 0.6 |
| | Post + 4 | 47.9 ± 14.8 | **3.2 ± 0.5** [a] | 5.6 ± 0.5 | 14.4 ± 0.6 | 2.5 ± 0.8 | 4.3 ± 0 | 13.7 ± 0.5 | 3.1 ± 0.2 | 0.6 ± 0.1 | 5.3 ± 0.2 | 0 ± 0 | 0.1 ± 0.1 | 49.5 ± 1.3 |



*CHYDR, carbohydrates with pentose and hexose subunits; PHLM, phenols and lignin monomers; LDIM, lignin dimers; LIPID, lipids,
alkanes, alkenes, bound fatty acids, and alkyl monoesters; ALKY, alkylaromatics; NCOMP, mainly heterocyclic N-containing compounds;
STEROL, sterols; PEPTI, peptides; SUBER, suberin; FATTY, free fatty acids.



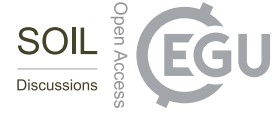

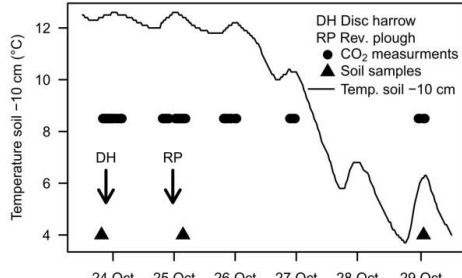



Figure 1. Timeline of the soil sampling and the $CO_2$ measurements in relation to the tillage
events. Additionally, soil temperature in 10 cm depth is plotted, recorded every 30 minutes
with an automated meteorological station (DALOS 535, F&C, Gülzow, Germany).




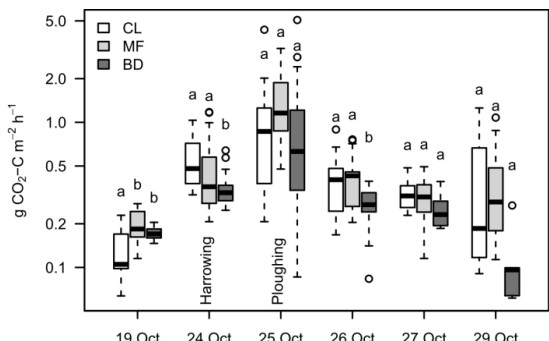



Figure 2. Soil $CO_2$ efflux around the days of tillage (harrowing up to 10 cm depth and
ploughing up to 30 cm depth). Note that for the days of tillage (24 and 25 October) only the
fluxes after tillage are included in order to get a better attribution of the tillage effect.
Different letters indicate significant differences ($p < 0.05$) in mean fluxes of the treatments
(CL, control; MF, mineral fertiliser; BD, biogas digestate) per each measurement day.





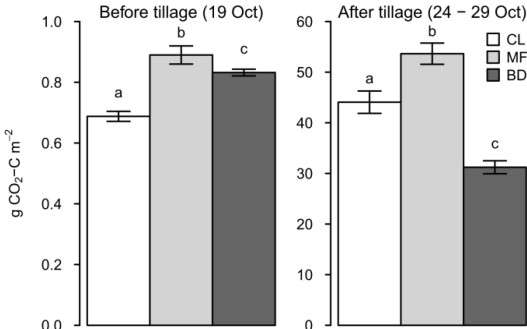



Figure 3. Cumulated soil $CO_2$ effluxes of a day before (19 October) and the period (24 – 29
October) after tillage. Different letters indicate significant differences in means of the
cumulated fluxes of the treatments (CL, control; MF, mineral fertiliser; BD, biogas digestate)
before and after, respectively. Error bars represent the standard deviation of interpolation by
bootstrapping after 250 iteration runs.




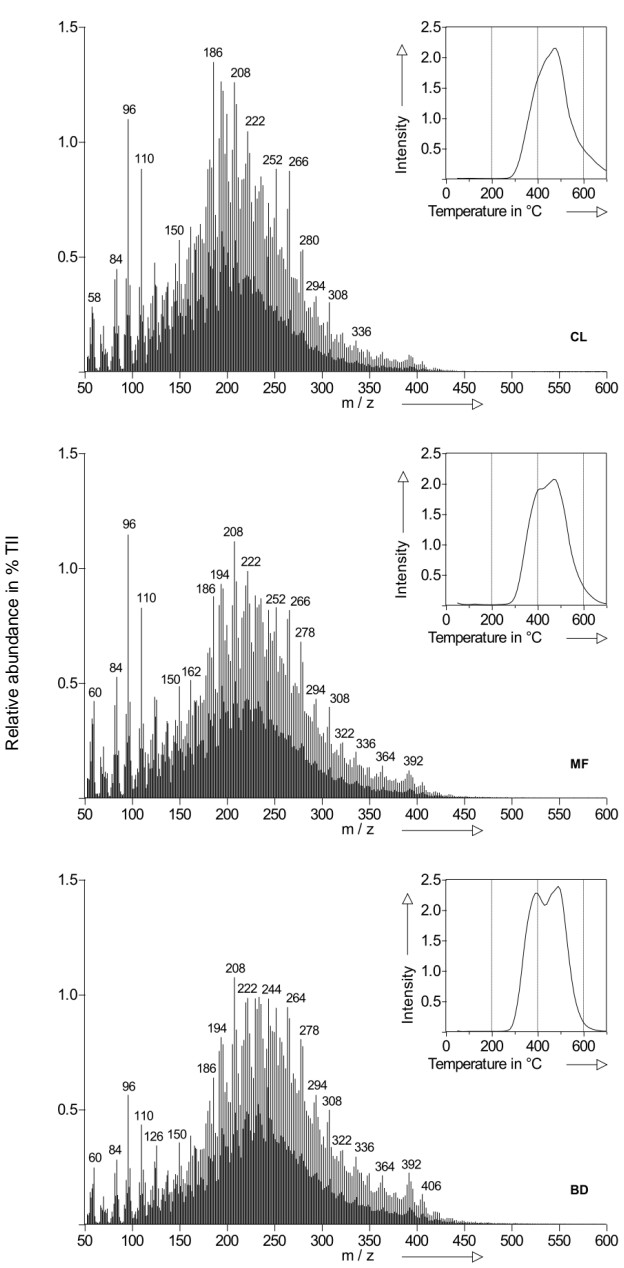


Figure 4. Thermograms of total ion intensity (TII, inserts upper right) and summed pyrolysis-
field ionisation mass spectra of the treatments (CL, control; MF, mineral fertiliser; BD, biogas
digestate) before tillage.





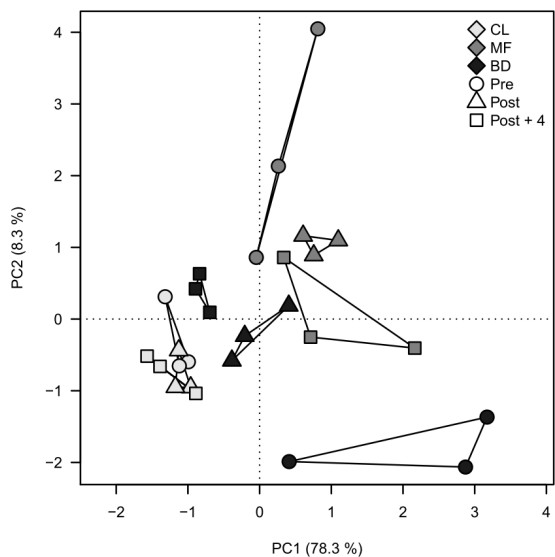



Figure 5. Principal component analysis of mass signals with significant differences according
to Wilks' $\lambda$. from the treatments (CL, control; MF, mineral fertiliser; BD, biogas digestate) of
the different sampling times (pre-tillage, post-tillage and post-tillage + 4 days). Treatments
and sampling times are depicted by different colours and symbols, respectively. Since the
areas integrated by the respective three sampling points did not overlap for the fertilised
treatments, a significant change of relative SOM composition can be assumed.




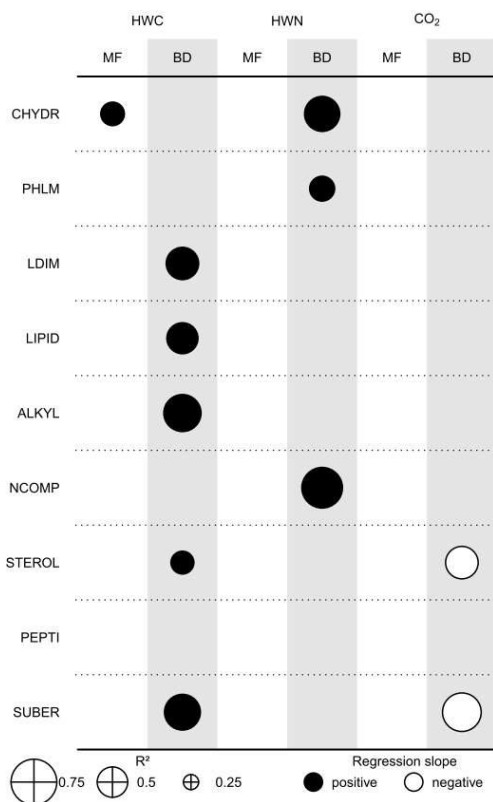



Figure 6. Significant ($p < 0.05$) linear correlations between absolute signal counts of the
compound classes and hot-water extractable carbon (HWC), hot-water extractable nitrogen
(HWN) and soil respiration ($CO_2$), respectively, with the corresponding coefficients of
determination ($R^2$) and direction of regression slopes, derived from the three soil sampling
dates.