# Peer review of "Tillage-induced short-term soil organic matter turnover"

_SOIL, 2015_

## Referee Comment (RC1) · S. Sleutel (Referee) · 16 Feb 2016

Soil-2015-91 presents a 'brave' (because challenging) attempt to finally better understand what exactly happens with specific OM constituents after soil tillage. The presented work is certainly complementary to a multitude of studies that have merely measured respiration or shifts in soil physical properties and environmental drivers. The similarity of mass spectra and thermograms of three different field objects really shows that the applied Py-FIMS technique's output is robust. And this renders it fit for detection of expectable small shifts in SOM composition in short term studies. The paper is well written, but I was not always convinced with explanations provided for a remarkable interactive effect of tillage and digestate application on soil $CO_2$ efflux. So in part it seems that some elements of the discussion need to be altered. But in general, the present paper is in good shape and only requires minor corrections. The

re-confirmation that sterols may inhibit microbial activity in soils really warrants further investigation. One suggestion for the data-analysis: to maximally explore linkages between (shifts in) the m/z datasets by tillage and (shifts in) $CO_2$ efflux, advantage could have been taken from newer statistical techniques like partial least squares regression. This would have resulted in principal components that maximally explain variation in $CO_2$ efflux (the dependent) and not in a set of components aimed at describing variation in the m/z data.

L 19-20 'Before tillage, BD showed much more volatilised matter (VM) during pyrolysis, indicating an increased amount of SOM.' Not really mention worthy in an abstract

42 suggest to replace by 'readily biodegradable OM' instead of 'labile organic matter', an ambiguous concept

52 'which SOC constituents that form the majority of SOM are mineralized' reads strangely, please rephrase

56 better: 'long-term stability in soil is still under debate. . .'

2.1 please provide sand, silt & clay percentages

115 so is this then a mouldboard plough?

143 all between the ( ) appear to be too much detail

161 'see 2.3', is written in 2.3, so where is this reference pointing at?

The purpose of measuring HWC and HWN should be better motivated in the introduction/M&M, when readers are mainly convinced that the whole intent of the present study was to understand which OM building blocks are preferentially degraded by consequences of tillage soil disturbance.

178 The reference is here part of your sentence and should not be expressed as (Leinweber et al., 2013) Leinweber et al. (2013Correct these mistakes in annotation throughout the tekst, several other examples follow later on in your text, e.g. 57 Möller
et al. (2015), 309, 320. . .)

Table 1 Designation of significant differences by the superscript letters in not clear. Is this the outcome of an ANOVA on all 9 treatment/date combinations? Seems that way, but be clearer in the caption text

Fig 3 Not clear what time intervals these cumulative $CO_2$ emissions represent. Should be indicated.

254 Indicate here in the text also the direction of 'differences' in the abundance of m/z 55, 60,etc.

282 presumably the authors are mentioning increases in the TII proportion of maker peaks for carbohydrates?

Fig 6 Why were correlations with the CL treatment's $CO_2$ efflux, HWC and HWN not displayed?

308 'BD'

350-353 A strange explanation: in the BD plots you also expect labile 'physically protected C' not contained within digestate to be equally present as in the MF plots, next to of course labile C in the amended digestate. So lifting of bio-availability of previously 'entrapped' labile OM should equally have occurred in the BD and MF treatments, equally contributing to a short-term $CO_2$ efflux. So I propose to reformulate this section. The microbial use efficiency theory which follows looks much more plausible. But then again, the HWC/HWN ratio was in fact much lower in all MF samples, and yet $CO_2$ efflux was higher. So in the end I think part 4.2 should end with an acknowledgement that the described mechanisms do not well explain why $CO_2$ efflux was lower after tillage in the BD when compared to the MF plots.

410 build-up

411 The link between trends of carbohydrates and heterocyclic N is not clear. This

statement requires further explanation.

401-409 Reads like a plausible explanation for the observed shifts in SOM biochemical composition. Can trends of individual m/z indicative for carbohydrates or peptides confirm that the short-term build-up of these substances is indeed through production of microbial biomass and metabolites?

771 suggest '…and time of tillage operations…'

780 indicate significance level

---

## Author Comment (AC1) · 18 Mar 2016

Dear Prof. Steven Sleutel,

Thank you for your comments – they are much appreciated because they acknowledge our efforts. Given below, we will enter your suggestions.

Beside them, we additionally described the exclusion of an outlier from the hot-water data in L 235-237:

'One of the replicates in MF exhibited exceptionally low HWC and HWN values. According to Dixon's Q-test, these values were outlier (one-third and half, respectively, as high as for the other replicates in MF) and thus excluded from further analysis.'

- One suggestion for the data-analysis: to maximally explore linkages between (shifts

in) the m/z datasets by tillage and (shifts in) CO2 efflux, advantage could have been taken from newer statistical techniques like partial least squares regression. This would have resulted in principal components that maximally explain variation in CO2 efflux (the dependent) and not in a set of components aimed at describing variation in the m/z data.

We seized your suggestion and applied PLSR to discriminate explanatory mass signals with regard to CO2. For visualisation, we used a PCA on these obtained masses, again:

'Partial least squares regression (PLSR) was used for discrimination (Barker and Rayens, 2003) to maximally explore linkages between shifts in the m/z data by tillage and shifts in CO2 efflux. PLSR models were built by using the R package "autopls" version 1.3 (Schmidtlein et al., 2015) with stepwise backward selection combined with a 10-fold cross-validation to substantially reduce the number of variables, i.e., to extract the variables with the highest explanatory power. The PLSR procedure was repeated 10.000 times to yield coherent results since the obtained PLSR models differed widely both in the number and in the choice of variables, thus in their predictive performance. Based on the performance index suggested by Bauwe et al. (2015), the 500 "best" models were extracted and, finally, the mass signals which were utilised more than 50 times in the latter models were chosen.' (220-230).

The outcomes are presented in the results and the discussion section:

'The discrimination of relative mass signals with PLSR to explain cumulated CO2 efflux revealed mainly functional groups from ketones and amides, peptides, carbohydrates as well as lignins and fatty acids (Table 3)' (308-311)

Table 3. Results of iterative partial least square regression for cumulated CO2 efflux as dependent variable and m/z data of all treatments and sampling times as explaining variables. m/z Molecule/compound class 17/18 Ammonium 31 [M+H]+ of formalde-hyde 34 H2S 43 C2H3O from ketones/amides and C3H7 propyl 46 Formic acid 55

C3H3O from ketones/amides 57 C3H5O from ketones/amdides and C4H9 butyl 73 Propanamide 83 C5H9N from peptides 85 C4H5O2 from carbohydrates 91 Fragment from peptides 98/99 Carbohydrates 206, 222, 230/231, 246, 254, 258 Lignins 296, 299, 337, 418, 424 Fatty acids (C19:1, C19:0, C22:2, C28:3, C28:0)

'Accordingly, these two m/z [17 and 18] were also selected by the PLSR as explanatory signals for CO2 efflux.' (425 -426)

'This suggestion is supported on the one hand by the effect of specific lignins on soil CO2 efflux (Tab. 3) since CO2 is an indicator for microbial decomposition activity (Kuzyakov, 2006).' (442-444)

- L 19-20 'Before tillage, BD showed much more volatilised matter (VM) during pyrolysis, indicating an increased amount of SOM.' Not really mention worthy in an abstract

We agree because there were no significant differences in means.

- 42 suggest to replace by 'readily biodegradable OM' instead of 'labile organic matter', an ambiguous concept

We changed the respective phrase to 'readily biodegradable (hereinafter referred to as "labile") organic matter' (now L 41-42) and also replaced 'labile' in the abstract (24).

- 52 'which SOC constituents that form the majority of SOM are mineralized' reads strangely, please rephrase

Rephrased to 'Admittedly, SOC accounts for the majority of SOM, but these correlations do not causally explain which SOC constituents are mineralised.' (51-53)

- 56 better: 'long-term stability in soil is still under debate. . .'

changed accordingly (55-56)

- 2.1 please provide sand, silt & clay percentages

sand, silt and clay percentages added (97)

- 115 so is this then a mouldboard plough?

It's a special type of mouldboard plough, yes. We stated the respective phrase more precisely to 'reversible mouldboard plough' (122) & also in L 256

- 143 all between the ( ) appear to be too much detail

We think that it is important to enable readers to comprehend the formula directly; therefore we discarded only specific values and would like to keep at least the respective units (144-147).

- 161 'see 2.3', is written in 2.3, so where is this reference pointing at?

The reference is pointing at 2.2; changed accordingly (168).

- The purpose of measuring HWC and HWN should be better motivated in the introduction/M&M, when readers are mainly convinced that the whole intent of the present study was to understand which OM building blocks are preferentially degraded by consequences of tillage soil disturbance.

The purpose of measuring HWC and HWN is now elucidated in the introduction after the outline about Py-FIMS: 'Hot-water extraction is a relatively simple method to release labile SOM and to estimate how much of soil C and N can be easily utilised my microorganisms (Leinweber et al., 1995). These labile pools have been suggested to be an important indicator of short-term changes in SOM quality due to soil management (Haynes, 2005). Furthermore, a significant proportion of hot water-extracted organic matter originates from microbial biomass. Thus, this approach is a potential indicator for changes in microbial biomass or activity (Sparling et al., 1998), which may reflect sources of $CO_2$ efflux following tillage. ' (80-87)

- 178 The reference is here part of your sentence and should not be expressed as (Leinweber et al., 2013) Leinweber et al. (2013 Correct these mistakes in annotation throughout the text, several other examples follow later on in your text, e.g. 57 Möller et al. (2015), 309, 320. . .)

Corrected accordingly.

- Table 1 Designation of significant differences by the superscript letters in not clear. Is this the outcome of an ANOVA on all 9 treatment/date combinations? Seems that way, but be clearer in the caption text

This is the outcome of Tukey's HSD, added to the affected caption texts.

- Fig 3 Not clear what time intervals these cumulative CO2 emissions represent. Should be indicated.

The intervals are now indicated: 'Cumulated soil CO2 effluxes on a day before (19 October, between 7 a.m. and 1 p.m.) and the period (24 October, 7 a.m. – 29 October, 1 p.m.) after tillage.'

- 254 Indicate here in the text also the direction of 'differences' in the abundance of m/z 55, 60,etc.

Direction of differences explained now: 'marker signals for carbohydrates and peptides (e.g., m/z 58, 60, 84, 69, 110, 126 and 162) were lower.' (275)

- 282 presumably the authors are mentioning increases in the TII proportion of maker peaks for carbohydrates?

We intended to mention changes in the proportions of compound classes – 'relative signal intensities' now corrected to 'relative proportions'. (303)

- Fig 6 Why were correlations with the CL treatment's CO2 efiňĆux, HWC and HWN not displayed? There were no significant correlations in treatment CL – a corresponding phrase is added to the caption, now.

- 308 'BD'

Corrected (333)

- 350-353 A strange explanation: in the BD plots you also expect labile 'physically

protected C' not contained within digestate to be equally present as in the MF plots, next to of course labile C in the amended digestate. So lifting of bio-availability of previously 'entrapped' labile OM should equally have occurred in the BD and MF treatments, equally contributing to a short-term $CO_2$ efflux. So I propose to reformulate this section. The microbial use efficiency theory which follows looks much more plausible. But then again, the HWC/HWN ratio was in fact much lower in all MF samples, and yet $CO_2$ efflux was higher. So in the end I think part 4.2 should end with an acknowledgement that the described mechanisms do not well explain why $CO_2$ efflux was lower after tillage in the BD when compared to the MF plots.

We found that 'even a single application of organic amendment can increase aggregate stability (Grandy et al., 2002) and thus the resilience against disruption by tillage might be promoted, leading to a better physical protection of labile soil C not contained within digestates.' (375-378)

&

'But in fact, the HWC/HWN ratio of BD after tillage was not lower than that of MF, so, in conclusion, the above described mechanisms do not well explain why the $CO_2$ efflux was lower after tillage in BD when compared to MF.' (393-395)

- 410 build-up

Corrected (448)

- 411 The link between trends of carbohydrates and heterocyclic N is not clear. This statement requires further explanation.

We added 'since a reduced C availability during the microbial transformation of N is suggested to promote formation of heterocyclic N instead of N immobilisation' (449-451).

- 401-409 Reads like a plausible explanation for the observed shifts in SOM biochemical composition. Can trends of individual m/z indicative for carbohydrates or peptides

coniňÄrm that the short-term build-up of these substances is indeed through production of microbial biomass and metabolites?

- We found an increase for muramic acid: 'This suggestion is supported on the one hand by the effect of specific lignins on soil $CO_2$ efflux (Tab. 3) since $CO_2$ is an indicator for microbial decomposition activity (Kuzyakov, 2006). On the other hand, a relative increase of the signals for m/z 125, 167, 185 and 203 was observed in the BD treatment (data not shown) which are assigned to the bacterial cell wall products N-acetylmuramic acid and N-acetylmuramyl-L-alanyl-D-isoglutamine (Bahr and Schulten, 1983)' (added L 442-448).

- 771 suggest '. . .and time of tillage operations. . .'

accepted (831)

- 780 indicate significance level

Implemented (834 & 842)

Please also note the supplement to this comment:
http://www.soil-discuss.net/soil-2015-91/soil-2015-91-AC1-supplement.zip

---

## Referee Comment (RC2) · Anonymous Referee #2 · 23 Mar 2016

Review of SOIL Discuss., doi:10.5194/soil-2015-81, 2016

Tillage-Induced short-term soil organic matter turnover and respiration

The MS addresses tillage-induced short-term $CO_2$ emissions and alterations of SOM under various fertilizer regimes by applying pyrolysis-field ionization mass spectrometry. The main problem with the MS is the lack of statistical replication of the fertilizer treatments; yet the authors refer extensively to statistical inferences, but based on tests with pseudo-replicates, which is generally considered as invalid. While the statistical design hampers valid comparisons between fertilizer treatments, it could be argued that effects of tillage operations within each fertilizer treatment could still be analysed, and the MS should be reworked with emphasis on this. At least, depending on journal standards, the authors should discuss the implication for the conclusions that comes

from using inferential statistics on pseudo-replicated data. The discussion at times is too speculative in relation to interaction between organic C (and N) pools and microorganisms (e.g., 423-437). Fewer points, with adequate support from the experimental data, should be focused on.

37: Rather: '...by the disruption of macroaggregates, leading to release of protected SOM...'

44: Remove author initials in references

52: The way the link between SOM and SOC is expressed sounds odd also in the reply to R1 ('Admittedly, SOC accounts for the majority of SOM, but...'). And maybe there is no need to introduce this issue here (but see Pribyl, 2010). To simplify, I would rather suggest to write: 'However, these correlations do not causally explain which organic components are mineralized.'

88: Although with some updates, this section (88-100) is closely following Fielder et al. (2015), and some condensation would be appropriate, e.g., replacing 91-95 by: "The top soil (0-30 cm) had an organic C content of $1.16 \pm 0.1\%$, pH of $7.4 \pm 0.9$, and bulk density of $1.51 \pm 0.08$ (mean $\pm$ standard deviation, n = 3) as measured according to Fiedler et al. (2015)." Also, for such an important parameter as pH for mineralisation, it would be reassuring (and easy) to provide more robust information than pH $7.4 \pm 0.9$. As a detail, always give same no. of decimals in mean and measure of variability (i.e., adjust $1.16 \pm 0.1\%$).

108: As the BD was applied to 160 kg N/ha in 2012, it would seem the assumption of 70% available N in total N was founded before the personal communication in 2014. The assumption is ok, but the writing could be made more consistent.

112: What is meant by 'in original matter'?

112: Suggest completing the chronological description: 'During the cropping season 2012, maize was grown according to conventional agricultural practice.' (and add if any

special management were implemented)

118: Give dimension (b x w) for collars

120: Maybe better: 'The adjacent collars were placed 1 m apart.'

128: Yes, even though acknowledged, pseudo-replication is of course a weakness of the study. And the HSD statistics as shown, e.g., in Table 1 are severely compromised by this design and should be reconsidered and preferably omitted

141: Suggest splitting sentence to improve readability: '... of a concentration measurement. This allowed discarding data...'

159: Does this literally mean 5-15 cm depth or is it 0-5 to 0-15 cm? Please specify. In the former case, why was the top-soil not considered?

218: For N only partly true and for HWN claim not statistically supported by data (even with faulty HSD)

166: Since 5 mg samples were used, a full description of procedures for homogenisation and subsampling should be given

191: What is meant by 'total nitrogen bound' ?

273: '...but it was not significant ($p < 0.01$).' Correct p value ?

298: rather '...in contrast to...'

299: '...to phenols, lignin...' or rather '...to phenols and lignin...' i.e., as individual compounds or as a group?

301: Correlation to free fatty acids not shown in Fig. 6 (free fatty acids not included in figure)

306: Don't abbreviate Table

306: See comment to 218

322: Speculative argument (322-323)

344: No need to invoke such ratios; suggest to delete 344-346.

348: Less available than what? You don't have a reference with the native biomass. Indeed this section is too speculative and should be shortened (347-365).

370: Leinweber et al. (2009) is not included in reference list

370: 'complementarily' – do you mean, e.g., 'collectively'

385: be specific: 'The VM increase...'

442: 'Since microbial activity can affects heterotrophic respiration...' this seems to be an understatement; delete 'can'

449: Again, this add to the many speculations put forward in the discussion (449-451)

455: I don't think the data has shown that lower $CO_2$ efflux was related to higher C use efficiency?

463: An awkward formulation: 'which likely inhibited an enhanced microbial activity.' Maybe use 'prevented' rather than 'inhibited'. Anyway the C/N ratios shown in Table 1 seem not to be critical for microbial mineralisation; rather N mobilisation would occur from mineralization.

Reference list is somewhat excessive (74 refs) with 44 unique references in the discussion alone, underlining the need for a more stringent focus in the discussion.

666: Ohkubu et al. (2012) not cited in text

560: add editor info

567: for consistency spell out journal names (also 590, 647, 565, 709, 736)

Table 1: Define in Table caption the meaning of pre, post and post+4

Table 2: As for Table 1

Figure 1: Spell out rev. plough in caption

Figure 2: '... (harrowing to ∼10 cm depth and ploughing to ∼30 cm depth)' '...in order to better visualize the tillage effect' Indicate specifically what boxes, error pars, points and symbols refer to in this case

Figure 6: This design is not helpful at all to the reader; please indicate R values numerically rather than by area-based symbols. And add explanation of acronyms.

---

## Author Comment (AC2) · 23 Mar 2016

Dear Prof. Sleutel, we added some minor improvements to the manuscript.

Line 19/20 'Before tillage, the Py-FIMS mass spectra revealed distinct differences in relative ion intensities of undisturbed soil compared to BD. . .' -> Changed to: 'Before tillage, the Py-FIMS mass spectra revealed distinct differences in relative ion intensities of MF and CL compared to BD' 73-75 'Furthermore, Py-FIMS of bulk SOM revealed alterations in laboratory incubation experiments and linked these to respiration and enzyme activities' -> 'Furthermore, Py-FIMS of bulk SOM revealed alterations in laboratory incubation experiments and allowed to link these to respiration and enzyme activities'

222 'PLSR models were built by using the R package "autopls" version 1.3' -> 'PLSR

models were built using function autopls of the R package "autopls" version 1.3'

229-230 'the 500 "best" models were extracted and, finally, the mass signals which were utilised more than 50 times in the latter models were chosen' -> 'the 500 "best" models were obtained and, finally, the mass signals which were utilised more than 50 times in the latter models were extracted'

474 'which also hints to a coupling' -> 'which also hints at a coupling'

482-484 'it is likely that increased amounts of sterols as they are typically found in biogas digestates (Leinweber, 2015, unpublished Py-FIMS data) delay the decomposition' -> 'increased amounts of sterols as they are typically found in biogas digestates (Leinweber, 2015, unpublished Py-FIMS data) likely delay the decomposition'

493-494 'CO2 efflux as an possible indicator for carbon use efficiency in terms of improved microbial N-availability decreased' -> 'lignins, ammonia and ammonium were discriminated as explanatory variables for cumulated CO2 efflux by PLSR'

496 'significant correlations among data sets from these methods' -> 'significant correlation exist among data sets from these methods'

Please also note the supplement to this comment:
http://www.soil-discuss.net/soil-2015-91/soil-2015-91-AC2-supplement.zip

---

## Author Comment (AC3) · 25 Apr 2016

Dear Madam, dear Sir,

Thank you for your careful reading – your detailed suggestions helped us to improve the MS considerably. You can find our following reply and the respective updated manuscript in the supplement-ZIP.

We acknowledge your justified statistical considerations about the replication of fertiliser treatments, but we do not agree with your suggestion to declare all testing between the respective treatments as invalid. Here is why:

The discussion about pseudo-replicates exists since there are ecological field studies. A true replicate in a statistical sense is independent of the other replicates for a factor

level, treatment, or whatever. When sampling in the field, the question arises "independent in regard to what"? Shall the climate be independent (because climate variation is a strong driver of GHG production and emission)? But then replicates would have to be placed many miles apart, which is rarely the case. Shall the soil be independent, i.e. should the measurements be carried out in randomized, not neighbouring blocks of a block design? But then, how to handle studies without treatments? Here, it is often assumed that the sampling locations are independent when they are placed far enough from each other. But how far is far enough? Well, that depends on the study site itself and for our study site we have good indication, that our sampling locations are independent enough to treat them as replicates because the soil of the study site shows very high small-scale variability well below the meter scale which tends to level off with increasing scale at the site level (Jacobs, 2014; LFA, personal communication, 2012). This is a strong argument for our results won't being any better or "more" valid if we had positioned the single sampling locations in greater distance from each other, or had spread them across gas sampling locations. Especially the relatively high standard deviances of our soil data corroborate this assumption (e.g., Py-FIMS data in Table 2). Furthermore, pseudo-replicated time series are common for soil gas measurements with chambers, since the bases for chamber deployment are anchored into soil to minimise soil disturbance (cf., Clough et al., 2015; Parkin and Venterea, 2010). Also, trace gas fluxes exhibit a high degree of spatial variability and it is, therefore, recommended to use rather chambers with large footprints (as ours) to integrate this variability (ibid.). In other words, the placement of our bases and their distances to one another were appropriate. According to Schank and Koehnle (2009), it is always necessary to consider the objectives of the study when discussing pseudo-replication. Our study aims at estimating gas fluxes ($CO_2$) from soil immediately after tillage to link them to short-term changes of SOM analysed by Py-FIMS. Based on this objective, we had very strong logistical reasons to carry out the sampling as described in the MS. First, the adequate coverage of these highly dynamic flux changes requires high frequency measurements in terms of minutes. This would have been hard to implement

with sampling locations sharing greater distances. Second, analyses by Py-FIMS are rather expensive (approx. 600€sample). Therefore, it was beyond our logistical and financial opportunities to gather data from all four replicates (plots) of each treatment. Thus, we decided to concentrate our resources and efforts on one plot of each treatment for these analyses. After all, pseudo-replication is only a cause for concern when the results are used to generalise outside the study system, which means that generalisability of pseudo-replicated studies may be lower than for those with true replication (Haddaway et al. 2014).

We acknowledge the pseudo-replication issue now in the methods and conclusion sections and argue as outlined above:

- 'Since the soil of the study site shows high small-scale variability (LFA, 2012, personal communication), sampling locations were expected to be independent enough to be treated as real replicates. Especially the relatively high standard deviances of our soil data corroborate this assumption.' (now 204-207)

- '. . .SOM composition of the investigated soil changed in the temporal scale of days. . .' (493-494)

We agree that the discussion about the link of carbon use efficiency to microbial activity may be too speculative and removed the related text, accordingly.

In the following, we answer to your specific comments. In parentheses at the end of our answers we give the lines in which the text is now.

37: Rather: '. . .by the disruption of macroaggregates, leading to release of protected SOM. . .'

- Changed accordingly (36)

44: Remove author initials in references

- We removed the author initials.

52: The way the link between SOM and SOC is expressed sounds odd also in the reply to R1 ('Admittedly, SOC accounts for the majority of SOM, but. . .'). And maybe there is no need to introduce this issue here (but see Pribyl, 2010). To simplify, I would rather suggest to write: 'However, these correlations do not causally explain which organic components are mineralized.'

- Suggestion accepted (50-51)

88: Although with some updates, this section (88-100) is closely following Fiedler et al. (2015), and some condensation would be appropriate, e.g., replacing 91-95 by: "The top soil (0-30 cm) had an organic C content of 1.16 $\pm$ 0.1%, pH of 7.4 $\pm$ 0.9, and bulk density of 1.51 $\pm$ 0.08 (mean $\pm$ standard deviation, n = 3) as measured according to Fiedler et al. (2015)." Also, for such an important parameter as pH for mineralisation, it would be reassuring (and easy) to provide more robust information than pH 7.4 $\pm$ 0.9. As a detail, always give same no. of decimals in mean and measure of variability (i.e., adjust 1.16 $\pm$ 0.1%).

- The text in lines 91-95 was replaced according to your suggestion: 'The top soil (0-30 cm) has an organic carbon content of 1.16% $\pm$ 0.10 (mean $\pm$ standard deviation, n = 3), pH of 7.4 $\pm$ 0.9 (n = 3) and bulk density of 1.51 g cm-3 $\pm$ 0.08 (n = 3), measured according to Fiedler et al. (2015).' (97-99)

- You are right, the pH should be measured preferably from every soil sample in studies like the present one. We will consider this in the future.

108: As the BD was applied to 160 kg N/ha in 2012, it would seem the assumption of 70% available N in total N was founded before the personal communication in 2014. The assumption is ok, but the writing could be made more consistent.

- Yes, the assumption was obviously founded before fertiliser application, corrected to 2012 (112).

112: What is meant by 'in original matter'?

[Figure]

- Thank you for mentioning this. We checked back again and found, that this was a wrong translation obviously. Instead, we meant 'undried material' and changed the text accordingly (116).

112: Suggest completing the chronological description: 'During the cropping season 2012, maize was grown according to conventional agricultural practice.' (and add if any special management were implemented)

- We changed the text according to your suggestion (118-119).

118: Give dimension (b x w) for collars

- 'The bases had dimensions of 79 x 79 cm' (127-128). - Additionally, we changed 'collars' to 'bases' throughout the manuscript.

120: Maybe better: 'The adjacent collars were placed 1 m apart.'

- We changed the text according to your suggestion but used "bases" instead "collars" (see above) (116-117).

128: Yes, even though acknowledged, pseudo-replication is of course a weakness of the study. And the HSD statistics as shown, e.g., in Table 1 are severely compromised by this design and should be reconsidered and preferably omitted

- As pointed out above, the placement of the bases and, hence, the pseudo-replicated time-series are common for trace gas measurements and in our view are appropriate in regard to spatial variability. That's why we think that the statistics are not compromised.

141: Suggest splitting sentence to improve readability: '... of a concentration measurement. This allowed discarding data...'

- Changed accordingly (146)

159: Does this literally mean 5-15 cm depth or is it 0-5 to 0-15 cm? Please specify. In the former case, why was the top-soil not considered?

- Thank you for bringing this up. We thought again about it and think that this information was an unjustified simplification. Here is why and our solution: Due to the uneven soil surface after tillage, it was not possible to sample accurately at 0 cm. Therefore, we decided to write "5 cm discarded" in the first place. After careful consideration, however, this does not reflect the real depth discarded. Therefore, we corrected the phrase and specified also the height of the used sample rings: '...were taken between 0 – 10 cm depth (depending on unevenness of soil surface due to tillage) with soil sample rings (h = 6.1 cm, V = 250 cm$^3$) ...' (164-165).

218: For N only partly true and for HWN claim not statistically supported by data (even with faulty HSD)

- Maybe this phrase was a bit ambiguous with regard to the comparison between treatments. Now, we point out the relation between MF and BD: 'Before tillage, the soil of all treatments had similar C and HWC contents, but differences appeared between MF and BD, where the N and HWN contents were slightly, though not significantly, higher in MF, resulting in significantly narrower C/N and HWC/HWN ratios in MF...' (242-245).

166: Since 5 mg samples were used, a full description of procedures for homogenisation and subsampling should be given

- We added more details according to your suggestion and the text now reads: '...the freeze-dried samples were finely ground and homogenized by a planetary ball mill. Then, about 2 g were transferred into a Petri dish with a spatula and three crucibles were filled by drawing them across. These subsamples of about 5 mg were thermally degraded...' (171-174).

191: What is meant by 'total nitrogen bound' ?

- This term refers to the sum of the organic and inorganic bound nitrogen, with the exception of elementary nitrogen. We adjusted the respective phrase: '...determination of hot-water extractable organic C (HWC) as well as of organic and inorganic bound N,

often referred to as "total nitrogen bound" (HWN). These measurements of organic C and total nitrogen bound are based. . .' (198-199).

273: '...but it was not significant (p < 0.01).' Correct p value ?

- Yes, indeed, another typo. Please, apologize. The correct value is p > 0.1 (298).

298: rather '. . .in contrast to. . .'

- Changed according to your suggestion (329)

299: '. . . to phenols, lignin. . .' or rather '. . .to phenols and lignin. . .' i.e., as individual compounds or as a group?

- No, not as individual compounds but as a group. We tried to make this more clear by editing the text to: '. . .to phenols and lignin monomers as well as to heterocyclic N-containing compounds. . .' (329).

301: Correlation to free fatty acids not shown in Fig. 6 (free fatty acids not included in figure)

- Thank you for your exact observation. We added the respective row to Fig. 6.

306: Don't abbreviate Table

- Changed accordingly

306: See comment to 218

- We changed the text and added information according to your suggestion: 'The C-, HWC-, N- and HWN-contents of the treatments showed no significant differences before tillage (Table 1), thus confirming the outcomes of other field experiments with similar fertilisers (Makádi et al., 2016; Odlare et al., 2014). However, the C- and N-contents obtained may not be representative for long-term effects of biogas digestate vs. mineral fertiliser which may also depend on soil texture (Makádi et al., 2016).' (337-342)

322: Speculative argument (322-323)

- Yes, we know, but we would like to discuss the issue anyway. However, we attenuated the argument by changing phrasing from 'most likely' to 'possibly' (351).

344: No need to invoke such ratios; suggest to delete 344-346.

- Deleted according to your suggestion.

348: Less available than what? You don't have a reference with the native biomass. Indeed this section is too speculative and should be shortened (347-365).

- Less available than undigested matter, which is generalizable (cf., Möller, 2015), and thus, there should be no need to provide a reference. We changed the phrase to make this clearer for the reader: '... likely less available to soil microorganisms than undigested organic matter, i. e., more "recalcitrant", since the most labile C is generally consumed in the biogas reactor ...' (374-376)

- In addition, we omitted the speculation about C use efficiency induced by N availability: 'As a consequence, the effect of increased $CO_2$ efflux after tillage as observed in CL and MF, may have been substantially reduced by a relative shortage of labile substrate for soil respiration in BD. The proportion of carbohydrates in BD derived from Py-FIMS, as discussed below, indicates not limited, but rather low levels of available C in the soil of BD.' (379-384)

370: Leinweber et al. (2009) is not included in reference list

- Thank you for this comment. We now added the reference and checked back through the whole reference list if all citations are in there.

370: 'complementarily' – do you mean, e.g., 'collectively'

- Yes. We changed the wording accordingly (389)

385: be specific: 'The VM increase. . .'

- Changed accordingly (403)

442: 'Since microbial activity can affect heterotrophic respiration. . .' this seems to be an understatement; delete 'can'

- We deleted "can" according to your suggestion (470)

449: Again, this add to the many speculations put forward in the discussion (449-451)

- We deleted the sentence according to your suggestion.

455: I don't think the data has shown that lower $CO_2$ efflux was related to higher C use efficiency?

- Yes, as discussed above, we removed the speculation about C use efficiency. However, since the outcome of the PLSR supports our suggestion of enhanced microbial N-turnover, we substituted the respective phrase by 'lignins, ammonia and ammonium were discriminated as explanatory variables for cumulated $CO_2$ efflux by PLSR' (480-481)

463: An awkward formulation: 'which likely inhibited an enhanced microbial activity.' Maybe use 'prevented' rather than 'inhibited'. Anyway the C/N ratios shown in Table 1 seem not to be critical for microbial mineralisation; rather N mobilisation would occur from mineralization.

- We replaced 'inhibited' by 'prevented': 'the absence of amendment may have led to a relative shortage of labile N as indicated by the higher HWC/HWN-ratio which possibly prevented an enhanced microbial activity.' (486-488)

- Though the C/N ratios might not be critical, the HWC/HWN could be "less optimal" in terms of availability of readily decomposable N. Recently, the availability was acknowledged as crucial for N use efficiency by Andresen et al. (2016). This is also discussed in 443-447, now: 'minor changes in SOM compounds in CL might be a consequence of the wider HWC/HWN ratio compared to MF and BD since it indicates a lower availability of labile N for microbial utilisation (Mengel, 1996). However, the total C/N ratios were not critical for microbial activity (Table 1) (Kuzyakov et al., 2000).'

Reference list is somewhat excessive (74 refs) with 44 unique references in the discussion alone, underlining the need for a more stringent focus in the discussion.

- We carefully went through all citations and omitted 27 unique references which provided no additional value (but also added a few during revision . . .).

666: Ohkubu et al. (2012) not cited in text

- In the process mentioned above we also deleted references that were left-overs from previous versions of the MS, like this one

560: add editor info

- Added accordingly (578)

567: for consistency spell out journal names (also 590, 647, 565, 709, 736)

- Changed accordingly

Table 1: Define in Table caption the meaning of pre, post and post+4

- Changed accordingly

Table 2: As for Table 1

- Same here

Figure 1: Spell out rev. plough in caption

- Changed to 'Mouldboard plough'

Figure 2: '. . . (harrowing to âĹij 10 cm depth and ploughing to âĹij 30 cm depth)' '. . . .in order to better visualize the tillage effect' Indicate specifically what boxes, error pars, points and symbols refer to in this case

- Boxes and symbols for fluxes on days after tillage are now 'distinguished by light grey backgrounds'

Figure 6: This design is not helpful at all to the reader; please indicate R values numerically rather than by area-based symbols. And add explanation of acronyms.

- R values are added

- For explanations, to retain flow of reading, we referred to Table 2: '(for explanation of abbreviations see Table 2)'.

References

Andresen, L. C., Björsne, A.-K., Bodé, S., Klemedtsson, L., Boeckx, P., and Rütting, T.: Depolymerization and mineralization - investigating N availability by a novel 15N tracing model, SOIL Discuss, 2016, 1–21, doi:10.5194/soil-2016-11, 2016.

Jacobs, O. (2014). Vergleich von zwei unterschiedlichen Kammersystemen zur Messung von bodenbürtigen Lachgasflüssen. Master thesis, University of Rostock

Clough, T. J., Rochette, P. Thomas, S. M., Pihlatie M., Christiansen, J. R., Thorman, R. E. (2015). Chamber design. In: de Klein C. A. M., Harvey M. J. (eds.) Nitrous Oxide Chamber Methodology Guidelines, Version 1.1 Ministry for Primary Industries, Wellington

Haddaway, N. R., Burden, A., Evans, C. D., Healey, J. R., Jones, D. L., Dalrymple, S. E., & Pullin, A. S. (2014). Evaluating effects of land management on greenhouse gas fluxes and carbon balances in boreo-temperate lowland peatland systems. Environmental Evidence, 3(1), 1.

Möller, K.: Effects of anaerobic digestion on soil carbon and nitrogen turnover, N emissions, and soil biological activity. A review: Agronomy for Sustainable Development, Agron. Sustain. Dev., 35, 1021–1041, doi:10.1007/s13593-015-0284-3, 2015.

Parkin, T. B. and Venterea, R. T. (2010). USDA-ARS GRACEnet project protocols,

chapter 3. Chamber-based trace gas flux measurements. Sampling Protocols. USDA-ARS, Fort Collins, CO, 3-1.

Schank, J. C., & Koehnle, T. J. (2009). Pseudoreplication is a pseudoproblem. Journal of Comparative Psychology, 123(4), 421.

Please also note the supplement to this comment:
http://www.soil-discuss.net/soil-2015-91/soil-2015-91-AC3-supplement.zip
* * *
[Figure]

|  | HWC | | HWN | | CO$_2$ | |
|  | MF | BD | MF | BD | MF | BD |
| CHYDR | 0.44 | | | 0.61 | | |
| PHLM | | | | 0.46 | | |
| LDIM | | 0.57 | | | | |
| LIPID | | 0.55 | | | | |
| ALKYL | | 0.64 | | | | |
| NCOMP | | | | 0.69 | | |
| STEROL | | 0.43 | | | | 0.40 |
| PEPTI | | | | | | |
| SUBER | | 0.62 | | | | 0.40 |
| FATTY | | | | 0.40 | | |

R²

Regression slope
● positive   ○ negative

**Fig. 1.** Figure 6

[Figure]

**Fig. 2.** Figure 2

---

## Author Response (AR1)

Dear Editor,

Thank you for commenting on our manuscript. Most of your comments are very well received and definitely help to improve the manuscript. However, we have the impression that you based your recommendations on an older version of the MS because many of the suggestions we already acknowledged and changed the MS accordingly. Below we provide detailed responses to all your comments including how we addressed it in the most recent version of the MS that is uploaded to the interactive discussion section (supplement to AC3).

Regarding the question of "pseudo-replicates" we provide detailed responses below. However, we want to express the following upfront: You know that most published work based on "cutting edge" methods with "objectively" very limited numbers of samples that can be run (NMR, mass spec, synchrotron) has almost no replications (neither true nor pseudo). In THIS context our data base is sound and there is no reason for us to withdraw any conclusion and we think that for a first time analysis with such an expensive analysis method, the sampling design is sufficient.

Thus, we look forward to a positive decision very soon.

Topical Editor's report:

Comments on "Tillage-induced short-term soil organic matter turnover and respiration" by S. R. Fiedler et al.

I agree with the comments of the two reviewers and in general also with the response of the authors. I would recommend major revisions of the manuscript before it might be considered for publication.

However, I have some further comments / suggestions for the revision of the manuscript:

I have similar problems as one of the reviewers with the statistical design of the study. The authors should be very clear that they used pseudo replicates. If I understood the paper correctly, the authors used just one experimental plot (6 x 30 m) and installed the three collars for CO2 measurements just in a distance of 1 m. This very small area was used to take the three soil samples too. This sampling strategy is not really representative for a field experiment also taking the spatial variability into account. The advantage of this strategy could be that small temporal changes might be detectable because of reducing the spatial variability.

- Our study's aim was to get a first insight into the short-term SOM turnover after tillage and, thus, into the temporal changes which, indeed, were obviously detectable. From this results we confirm that Py-FIMS is an appropriate method to study short term SOM turnover in soils. In the future researchers should try to address spatial variability in more detail.
- In accordance with that we stated in our reply to referee #2, "our study aims at the estimation of gas fluxes ($CO_2$) from soil immediately after tillage to link them to short-term changes of SOM analysed by Py-FIMS". Additionally, "the placement of our bases and their distances to one another were appropriate" since they integrate the high spatial variability of gas fluxes from soil (cf., Clough et al., 2015; Parkin and Venterea, 2010).
- The three collars and the distances between them had a circumference of at least 4.4 x 0.8 m in each treatment-plot. The soil samples were taken from outside of this area which means

that these samples integrated over an area of about 2.5 m², which may be on the edge, but was necessary to ensure the link to the gas fluxes.

- Additionally, we now refer to the master thesis by Jacobs (2014) who demonstrated that $N_2O$ fluxes from the soil of the study site show very high small-scale variability well below the meter scale which tends to level off with increasing scale at the plot level: "Although the relatively small sampling areas around the bases in each treatment plot might suggest a 'pseudo-replication' in soil sampling, we have evidence suggesting a very high spatial variability in the soil, which alleviates this problem: In a master thesis on spatial variability, Jacobs (2014) revealed that N2O fluxes from the soil of the study site show very high small-scale variability well below the meter scale. Therefore, we assume 'real', i.e. independent replicates…" (Lines 466-470)

The observed differences were mostly quite small and just of few of them were statistically significant (particularly differences observed in the relative proportions of compound classes - table 2). Therefore, I recommend to adapt the whole discussion section taking the limitations of the design and the rather small differences into account.

- We acknowledge that the differences were often rather small. But concurrently, the detection of small differences is a specific feature and advantage of Py-FIMS. However, we added a section ("4.4 Limitations") concerning the observed small differences between the plots, which is also pointing at the reasonable detection of significant temporal changes: "though the comparison *between* the treatments should be done carefully because of possibly rather small differences. At the same time, due to the, thus, potentially lowered influence of spatial variability, our sampling design might have biased our results towards the detection of even small temporal changes *within* the treatments. Because we are mainly interested in the impact of tillage, this limitation is not interfering with our findings " (465-476)

The paper would really profit from shortening.

- You are right. The initial version of the MS was quite lengthy. However, already the now actual version on SOILD is substantially shorter due to edits in response of the comments of the two reviewers. In addition we carefully went through the MS again and tried to shorten it further, whereby also addressing your above raised point that we should adapt the whole discussion to the "limitations" of the study.

Some further comments:

Abstract:
Lines 19-20: I do not think that a higher proportion of volatilized matter during pyrolysis indicates an increased amount of SOM.

- You are right, additionally to higher amounts of SOM, increased proportions of volatised matter may also indicate a lower stability of SOM. However, the respective phrase was erased from the abstract in a later version of the manuscript. The context of volatised matter and SOM amount/stability is discussed in lines 385 ff.

Line 26: the high spatial variability is based on a personal communication – very weak base. I wouldn't use such an argument in the abstract.

- We agree that this phrase has no additional value for the abstract, therefore we removed it. However, the claim about the spatial variability is corroborated by own investigations as stated above.

Lines 27-28: increased in comparison to ???

- It increased in comparison to before tillage. This information is given in the previous sentence: "Significant changes in SOM composition were observed following tillage. In particular, … increased proportions of N-containing compounds in BD".

Material and methods:
Lines 91-92: The content of organic C in the soil is much higher than the reported ones in table 1. This difference was not considered in the discussion. Are there any reasons for that?

- Yes, there are. This was the amount reported from our previous paper. The sampling locations are ~50 m away from the spots we used here. We corrected this value to the averaged „Pre"-values from table 1 to not leave the reader with different numbers that cannot be understood without context. The values for bulk density and pH were obtained from the present spots before tillage.

I did not read any information about the history of the treatments. What kind of management has been applied before starting the experiment? Was the BD just once applied?

- Yes, BD was applied only once, because this trial was established for a single year. In the history, there were other trials at the site with winter wheat followed by maize. We edited the respective section to better reflect those details: "Before our study period, during other trials, winter wheat (*Triticum aestivum* L.) followed by maize were grown on the field. (102-103). "The BD for this single application originated from …" (112/113)

Lines 159-160: Please describe how samples were taken – composite samples of xy cores / subsamples??

- The samples were taken directly with the sample rings: "Three replicates of bulk soil samples were taken between 0 – 10 cm depth (depending on unevenness of soil surface due to tillage) directly with three soil sample rings (h = 6.1 cm, V = 250 cm³)" (163-165) and were "… fixed immediately with liquid nitrogen and splitted thereafter into subsamples for freeze-drying and for oven-drying at 60° C." (163-170)

Lines 175-176: I do not understand the implications / functions of the averaged survey spectrum per

treatment. Does it mean that the proportions of compound classes were calculated based on one spectrum per treatment? How did you calculate means and standard deviations?

- The averaged survey spectrum refers to the figure that we show. We redescribed the respective phrase: "For plotting, the three replicates of each sample were then averaged to one final survey spectrum" (181-182). For analysis we considered the replicate runs separately as replicates.

As one of the reviewers I did not understand the purpose of the hot water extraction.

The purpose of measuring HWC and HWN is now elucidated in the introduction after the outline about Py-FIMS: "Hot-water extraction is a relatively simple method to release labile SOM and to estimate how much of soil C and N can be easily utilised my microorganisms (Leinweber et al., 1995). These labile pools have been suggested to be an important indicator of short-term changes in SOM quality due to soil management (Haynes, 2005). Furthermore, a significant proportion of hot water-extracted organic matter originates from microbial biomass. Thus, this approach is a potential indicator for changes in microbial biomass or activity (Sparling et al., 1998), which may reflect sources of $CO_2$ efflux following tillage. (77-84)

Discussion

In the discussion I would prefer a more condensed way to discuss the most important findings of the study in a more straightforward way.
- This was acknowledged by the other referees, too, and, therefore, the recent MS should be already more concentrated. However, we have re-inspected the discussion based on your suggestions below and tried to tighten it even more.

I got the impression that OM in the digestate treatment was sometimes more and sometimes less available to microorganisms. That is not very consistent.

- Indeed there are two elements to the story of OM availability in the digestate treatment: On the one hand there is a generally increased availability of stable OM fractions from BD residues after tillage. On the other hand, there is a relative shortage of available C. For this reason, we refined the section on $CO_2$ which might have been confusing in this context: "… C originating from the digestates is likely less available to soil microorganisms compared to undigested organic matter …" (361-362). The discussion about Py-FIMS is following this differentiation quite consistently.

The discussion from lines 350-365 is not convincing. I would think that OM from BD might enhance soil aggregation, resulting in less disruption of aggregates during tillage.

- The lines you are referring to were already rearranged and mostly omitted in the most recent version of the MS, because the discussion about carbon use efficiency was too speculative. A short discussion on a higher resilience against disruption of macroaggregates due the use of an organic amendment has been introduced already: "On the other hand, even a single application of organic amendment can increase aggregate stability (Grandy et al., 2002).

Therefore, the resilience against disruption by tillage might be promoted, leading to a better physical protection of labile soil C not contained within digestates" (364-367).

Line 381-385: I do not see that, i.e. an enrichment of (stable) SOM indicated by a higher portion of VM.

- Here, we disagree. The share of VM as well as the total ion intensity are widely accepted as indicators for the amount and stability of SOM and SOC, respectively (Sorge et al., 1993; Wilcken et al., 1997; Leinweber and Schulten, 1995). The mean values of VM before tillage were 5.2% (BD), 3.9% (MF) and 3.6% (CL). To corroborate our position, we introduced also TII into this section. TII was 44.3 (BD), 34.2 (MF) and 41.5 (CL) x $10^6$ counts mg$^{-1}$. Though we were not able to test for significant differences between the treatments, BD had the highest mean values for both variables. That's why we "suggest a tendency to elevated SOM" (387). Maybe the attribution of organic matter from BD application as "rather stable" causes confusing in this context. Thus, we omitted it and the respective section reads now: "VM as well as TII, which are indicators of SOM content (Sorge et al., 1993) and also of its stability (Ludwig et al., 2015), were larger in BD than in MF and CL before tillage (Table 2). This suggests a tendency to elevated SOM due to application of organic matter with biogas digestate." (385-387).

The discussion related to the rather small changes in hot water extractable N (and C) and relationships to Py-FIMS data is highly speculative and not very convincing. I would recommend to shorten this part substantially.

- We reconsidered this part and agree partly about the speculative character due to the rather small changes. We removed Figure 6 and most text bits about the single correlations between hot-water extracts and the compound classes in the results and the discussion sections, but we kept the correlations that were previously described by other authors, i.e., the relations between HWC and carbohydrates as well as between $CO_2$ and sterols. (312-323 and 435-444)

I would strengthen the discussion related to relations between CO2 effluxes and changes in SOM composition (starting from line 438).

- We followed your suggestion and strengthened this part. (445-452)

In the whole discussion about labile N I missed information about mineral N in the soil in the different treatments. This is the most available form of N.

- The signals of $m/z$ 17 and 18 ($NH_3$ and $NH_4$) are discussed in the recent MS version (395-399) because we included it before in reaction to reviewer comments.

Tables and figures

The standard deviations as indicated in table 1 are quite low. That might be the result of the design of the study and does not support the argument of high spatial variability.

- In Table 1 that's true, but SD in Py-FIMS data (Table 2) are rather high and the latter method is much more sensitive to differences in compound classes than simple C and N determinations. However, after all, the MS is now focussed on turnover processes rather than on differences between fertilisation regimes.

Independently from your comments, we have added an additional reference supporting the idea that microorganisms utilise lignin-derived moieties at low initial availability of carbohydrates: "Recently, Rinkes et al. (2016) also found that decomposers may break down lignin to acquire C for their metabolism in the absence of available labile C. (415-417)", added with a respective phrases in lines 453-460: "…enhanced microbial N-turnover by tillage in soils amended with biogas digestates; possible co-occurring with the decomposition of lignin as C source due to a relative shortage of carbohydrates. […] increase of N-containing compounds along with decomposition of lignins and formation of carbohydrates and peptides. "

References

Clough, T. J., Rochette, P.  Thomas, S. M., Pihlatie M., Christiansen, J. R., Thorman, R. E. (2015). Chamber design. In: de Klein C. A. M., Harvey M. J. (eds.) Nitrous Oxide Chamber Methodology Guidelines, Version 1.1 Ministry for Primary Industries, Wellington

Jacobs, O. (2014). Vergleich von zwei unterschiedlichen Kammersystemen zur Messung von bodenbürtigen Lachgasflüssen.  Master thesis, University of Rostock

Leinweber, P. and Schulten, H. R.: Composition, stability and turnover of soil organic matter: investigations by off-line pyrolysis and direct pyrolysis-mass spectrometry, Journal of Analytical and Applied Pyrolysis, 32, 91–110, doi:10.1016/0165-2370(94)00832-L, 1995.

Parkin, T. B. and Venterea, R. T. (2010). USDA-ARS GRACEnet project protocols, chapter 3. Chamber-based trace gas flux measurements. Sampling Protocols. USDA-ARS, Fort Collins, CO, 3-1.

Sorge, C., Müller, R., Leinweber, P., and Schulten, H.-R.: Pyrolysis-mass spectrometry of whole soils, soil particle-size fractions, litter materials and humic substances: statistical evaluation of sample weight, residue, volatilized matter and total ion intensity, Fresenius' Journal of Analytical Chemistry, 346, 697–703, doi:10.1007/BF00321275, 1993.

Wilcken, H., Sorge, C., and Schulten, H. R.: Molecular composition and chemometric differentiation and classification of soil organic matter in Podzol B-horizons, Geoderma, 76, 193–219, doi:10.1016/S0016-7061(96)00107-3, 1997.

---

## Author Response (AR2)

Dear Karsten, thank you for the good news. We followed your suggestion and removed the respective parts. Additionally, lines 395 to 399 have been moved to lines 418 to 422; and the phrase "due to application of organic matter with biogas digestate" in line 387 has been changed to "through application of BD."

Thank you for your time and expertise.

Best regards,
Sebastian

[revised manuscript text omitted]